# Human T cells engineered with a leukemia lipid-specific TCR enables donor-unrestricted recognition of CD1c-expressing leukemia

Michela Consonni [1,2], Claudio Garavaglia[1], Andrea Grilli [3], Claudia de Lalla[1], Alessandra Mancino[1], Lucia Mori [4], Gennaro De Libero[4], Daniela Montagna[5], Monica Casucci[6], Marta Serafini[7], Chiara Bonini [8], Daniel Häussinger [9], Fabio Ciceri[10], Massimo Bernardi [10], Sara Mastaglio[10], Silvio Bicciato [3], Paolo Dellabona [1✉] & Giulia Casorati [1✉]

Acute leukemia relapsing after chemotherapy plus allogeneic hematopoietic stem cell transplantation can be treated with donor-derived T cells, but this is hampered by the need for donor/recipient MHC-matching and often results in graft-versus-host disease, prompting the search for new donor-unrestricted strategies targeting malignant cells. Leukemia blasts express CD1c antigen-presenting molecules, which are identical in all individuals and expressed only by mature leukocytes, and are recognized by T cell clones specific for the CD1c-restricted leukemia-associated methyl-lysophosphatidic acid (mLPA) lipid antigen. Here, we show that human T cells engineered to express an mLPA-specific TCR, target diverse CD1c-expressing leukemia blasts in vitro and significantly delay the progression of three models of leukemia xenograft in NSG mice, an effect that is boosted by mLPA-cellular immunization. These results highlight a strategy to redirect T cells against leukemia via transfer of a lipid-specific TCR that could be used across MHC barriers with reduced risk of graft-versus-host disease.

---

[1] Experimental Immunology Unit, Division of Immunology, Transplantation and Infectious Diseases, IRCCS San Raffaele Scientific Institute, Milan 20132, Italy. [2] Vita-Salute San Raffaele University, Milan, Italy. [3] Department of Life Sciences, University of Modena and Reggio Emilia, Modena, Italy. [4] Experimental Immunology, Department of Biomedicine, University of Basel and University Hospital, Basel, Switzerland. [5] Foundation IRCCS Policlinico San Matteo; Department of Sciences Clinic-Surgical, Diagnostic and Pediatric, University of Pavia, Pavia, Italy. [6] Innovative Immunotherapies Unit, Division of Immunology, Transplantation and Infectious Diseases, IRCCS San Raffaele Scientific Institute, Milan, Italy. [7] M. Tettamanti Research Center, University of Milano-Bicocca, Monza, Italy. [8] Experimental Hematology Unit, Division of Immunology, Transplantation and Infectious Diseases, IRCCS San Raffaele Scientific Institute, Milan, Italy. [9] NMR-Laboratory, Department of Chemistry, University of Basel, Basel, Switzerland. [10] Hematology and Bone Marrow Transplant Unit, IRCCS San Raffaele Scientific Institute, Milan, Italy. ✉email: dellabona.paolo@hsr.it; casorati.giulia@hsr.it

Acute leukemia is a heterogeneous group of hematological cancers that all involve malignant proliferation of immature hematopoietic cells which invade the bone marrow, blood, and extramedullary sites, and can affect a wide range of age groups with varying degrees of severity. The most common adult and pediatric types are acute myeloid leukemia (AML) and acute lymphoblastic leukemia (ALL), respectively, with an incidence of about 4 cases/100,000 in the western world[1]. Current treatment for low-risk acute leukemia consists of several courses of chemotherapy; whereas intermediate/high-risk diseases need subsequent consolidation with allogeneic hematopoietic stem cell transplantation (HSCT). Although many acute leukemia patients achieve complete remission, approximately 20–30% eventually undergo disease relapses. These patients have a very poor prognosis, and effective treatments targeting the post-transplant recurrence of residual leukemia blasts is a major unmet clinical need[2]. One approach is the transfer of allogeneic donor-derived T cells, which may induce a beneficial graft-versus-leukemia (GvL) reaction capable of maintaining remission[3]. However, the grafted allogeneic T cells can also attack patients' non-hematopoietic tissues, inducing detrimental graft-versus-host disease (GvHD)[4]. To avoid this risk it is necessary to develop a T cell-mediated therapeutic strategy targeting malignant cells, while maintaining hematopoietic capacity among grafted cells and preserving organ functions in recipient patients[3]. T cells specific for tumor-derived peptide antigens presented by MHC molecules have a critical role in controlling cancer progression[5], and ex vivo engineering of T cells with MHC-restricted T cell receptors (TCR) specific for peptides derived from tumor antigens has the potential to generate large numbers of tumor-specific effector T cells from a patient[5]. However, because of the high polymorphism of MHC, this approach requires the isolation of many different tumor-specific TCRs restricted to the most frequent HLA alleles.

An alternative strategy is T cell-mediated targeting of tumor antigens presented by monomorphic molecules, such as CD1 and MR1. CD1 molecules are MHC class I-related molecules specialized in presenting lipid antigens[6], and humans express five CD1 genes: CD1a, CD1b and CD1c (group 1); CD1d (group 2); and CD1e (group 3). CD1-restricted T cells recognize foreign lipids from pathogenic bacteria[7], but are also strongly reactive against specific cell-endogenous "self" lipids[7]. These CD1 self-reactive T cells constitute 0.1 to 10% of circulating T lymphocytes[8,9], and their role in the immune response is incompletely understood. Group 1 CD1 molecules are expressed only by mature hematopoietic cells[6], and their expression is elevated in leukemia and lymphoma cells; indeed, CD1c is detected on all the blasts of 54% of adult and 45% of pediatric AML patients, and on all the blasts of 71% of adult and 26% of pediatric B cell acute lymphoblastic leukemia[10]. Furthermore, a group of CD1c self-reactive T cell clones that recognizes methyl-lysophosphatidic acid (mLPA), a novel lipid antigen that is highly enriched in leukemia cells compared to healthy leukocytes has been identified[10]; these T cells killed leukemia blasts in vitro and in immunodeficient mouse xenograft models[10]. These data led us to investigate whether generating CD1c-retargeted allogeneic T cells recognizing mLPA expressed by leukemia cells might represent a novel immunotherapy able to overcome the limitation of conventional MHC-restricted T cell therapies for leukemia.

CD1c-restricted T cells specific for mLPA are potentially ideal candidates for the adoptive immunotherapy of acute leukemia due to the following unique characteristics: (i) CD1c is not polymorphic[6], allowing the generation of donor-unrestricted (universal) effector T cells; (ii) CD1c is not expressed by parenchymatous organs[6], minimizing the risk of eliciting GvHD; (iii)

mLPA is highly enriched in malignant cells, but not in normal cells[10]; and (iv) mLPA belongs to lipid species participating in the oncogenic process and tumor progression[10,11] that, being produced through complex synthetic pathways, may also be less susceptible than protein antigens to immune-mediated loss caused by single gene mutation.

In this work, we transduce a range of mLPA-specific TCRs into human T cell lines and primary human T cells in order to identify a lead candidate for preclinical testing. We show that transduction with the selected lead mLPA-specific TCR efficiently retargeted polyclonal T cells from a range of donors against CD1c-expressing acute leukemia cells in vitro, while sparing healthy CD1c-expressing circulating monocytes, Dendritic Cells (DC) and B cells. When we transfer these TCR-expressing T cells into acute leukemia xenograft mouse models we show evidence of tumor infiltration, inhibited tumor growth, and prolonged survival.

## Results

**CD1c is expressed in AML, B-ALL and DLBCL.** CD1c expression on blasts from two cohorts of pediatric and adult acute leukemia patients had been previously detected by flow cytometry[10]. To confirm and extend these original findings, we first established the expression levels of all group 1 and 2 CD1 genes in acute leukemia by interrogating 16 publicly available gene expression datasets from adult and pediatric AML, B-ALL, and T cell acute lymphoblastic leukemia (T-ALL) patients (Supplementary Table 1). These datasets were filtered according to uniformity of platforms, e.g. Affymetrix, completeness of available clinical information and with adequate number of samples for each leukemia type. This analysis revealed that CD1c was the most abundantly expressed member of the CD1 family in both AML and B-ALL (Fig. 1a, Supplementary Fig. 1a), confirming our previous findings. The GSE18497 dataset, the only one available with matched diagnosis-relapse paired samples, reported comparable levels of CD1 gene expression in B-ALL and T-ALL at diagnosis and at relapse (Fig. 1b), supporting the immunological targeting of CD1 molecules at both early and late stages of these malignancies. When we extended our analysis to evaluating CD1c expression in a broad range of tumor types using the Cancer Genome Atlas (TCGA), we also found significant overexpression in Diffuse Large B Cell Lymphoma (DLBCL) compared to normal tissue (Fig. 1c, Supplementary Table 2). Thus, CD1c-reactive T cells have the potential to target AML, B-ALL, T-ALL, and DLBC lymphoma expressing CD1c.

**Selection of optimal CD1c tumor-reactive TCR for leukemia targeting.** Having established that CD1c was abundantly expressed in acute leukemias and DLBCL, the next step was to identify the optimal CD1c-restricted TCR to retarget T cells against these malignancies. We therefore selected five CD1c self-reactive T cell clones[10] and sequenced their TCR α and β genes: each T cell clone expressed a unique TCR, some sharing V regions, but always with diverse junctional sequences (Supplementary Fig. 1b). We then cloned the cDNA sequence encoding the TCR Vα and Vβ chains of each TCR into a lentiviral vector (LV) (Supplementary Fig. 1c). The human Vα and Vβ regions were joined to the respective mouse C regions (as described[12]) to avoid mispairing of the transduced TCR with the endogenous chains in the transduced T cells, and to increase the expression of the transduced TCR. We selected Jurkat 76 cells as the transduction target because they do not express an endogenous TCR[13]; however, these cells normally express all CD1 molecules, so to avoid any CD1c-dependent homotypic recognition by the TCR-transduced cells, we deleted the B2M gene with CRISPR/Cas9

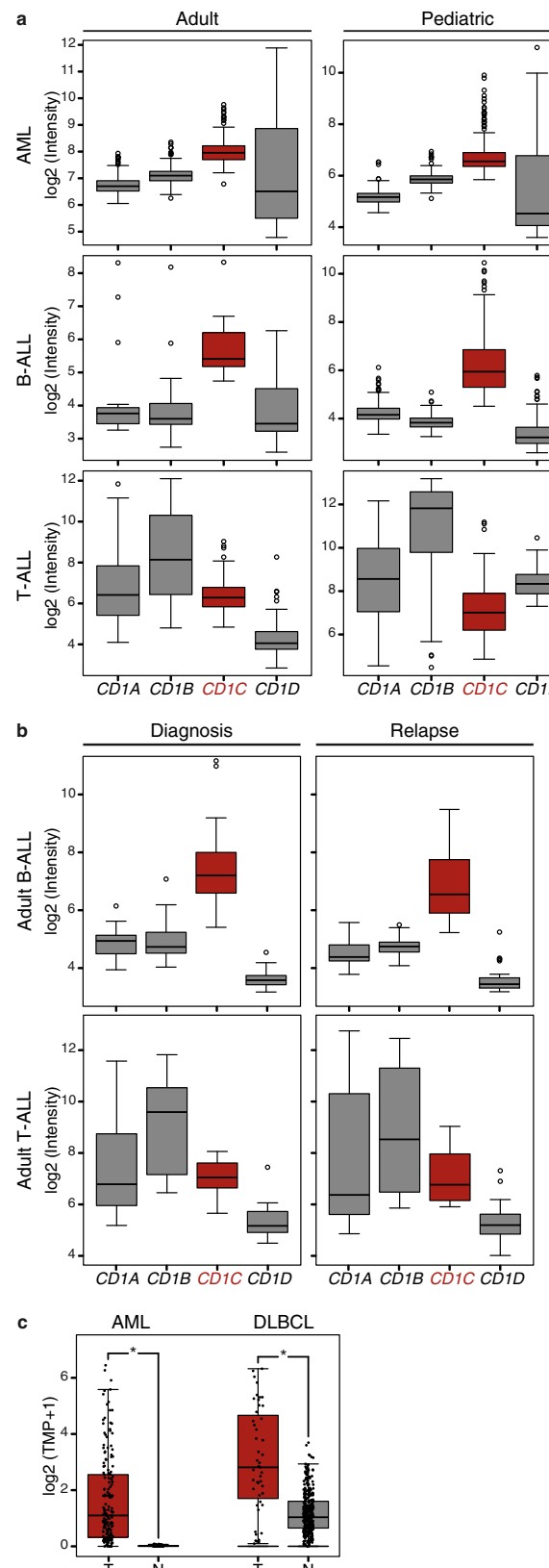

**Fig. 1 CD1 gene expression in leukemia datasets.** Analysis of group 1 and group 2 CD1 expression levels in different types of leukemia from published datasets highlights higher and more uniform expression of CD1c in both adult and pediatric Acute Myeloid Leukemia (AML) and B-Acute Lymphoblastic Leukemia (B-ALL). **a** Expression level of *CD1* genes across a selection of leukemia datasets [adult AML: GSE12417; pediatric AML: GSE17855; adult B-ALL: GSE14834; pediatric B-ALL: GSE13576; adult T-Acute Lymphoblastic Leukemia (T-ALL): GSE14618; pediatric T-ALL GSE50999; the expression levels of *CD1* genes in additional datasets are reported in Supplementary Fig. 1a]. Sample size of each dataset is detailed in Supplementary Table 1. **b** Elevated expression of CD1c in adult B-ALL is preserved in patients not responding to therapies, while its expression remains low in T-ALL at the time of diagnosis and at relapse. Data were obtained from GSE18497 leukemia dataset. Sample size of the dataset is detailed in Supplementary Table 1. **c** CD1c expression in TCGA-AML and TCGA-Diffuse Large B Cell Lymphoma (DLBCL) tumor samples (T) compared to their corresponding normal tissues (N). The analysis included 173 AMLs and 70 normal bone marrow samples, and 47 DLBCLs and 337 normal blood samples (Table S2). Data on tumor samples were obtained from TCGA, while normal sample data were obtained from GTEx. CD1c expression was considered significant at $^*P < 1.00\text{E}{-}10$ in a one-way ANOVA. In the box plots, the box is enclosed between the first quartile (25th percentile) and the third quartile (75th percentile), the center is represented by the median, the whiskers are defined by 1.5 times the interquartile range (i.e., it is the distance between the upper and lower quartiles).

We next asked whether TCR-transduced Jurkat 76 β2m⁻ (TCR-JK) cells could recognize AML cell lines engineered to express CD1c (Fig. 2b), and loaded or not loaded with synthetic mLPA. The DN4.99, PZP8A6, and DN4.2 TCR-JK cells expressed significantly higher levels of the activation marker CD69 following co-culture with CD1c-expressing THP-1 (Fig. 2c) or K562 (Fig. 2d) cells compared to WT, and their activation was further significantly increased in the presence of mLPA. Although the P8E3 TCR was detected at the JK cell surface (Fig. 2a), P8E3 TCR-JK cells did not show activation in response to either AML cell line (Fig. 2c, d); interestingly, the P8E3 TCR shares the same Vα/Vβ genes as DN4.99 TCR (Supplementary Fig. 1b), but has different rearrangements, suggesting that the TCR CDR3 regions are important for recognition of mLPA. When we compared the reactivity of the DN4.99, PZP8A6 and DN4.2 TCR-JK cells against different doses of mLPA presented on K562-CD1c cells we found that, despite very similar levels of surface expression, the DN4.99 TCR exhibited a lower EC₅₀ (half-maximal effective concentration) than the other two TCRs, and the highest CD69 expression (Fig. 2e), suggesting stronger functional avidity[14,15] upon antigen engagement.

Next, we transduced the chimeric DN4.99, PZP8A6, and DN4.2 TCRs into freshly isolated circulating T cells from healthy donors ($n = 3$–5) and measured their expression and ability to recognize mLPA. All three TCRs were expressed on at least 70% of transduced T cells (Fig. 3a), but DN4.99 TCR induced the strongest and most sustained (up to 27 days post-transduction) downregulation of endogenous TCRs (Fig. 3b, Supplementary Fig. 2a). This is consistent with published reports showing a higher affinity of mouse TCR Cα and Cβ regions for the human CD3ζ compared to the human C regions[12], resulting in the sequestration of available CD3ζ molecules, which are the rate-limiting components for the transport of all TCR/CD3 complexes to the cell membrane[16,17] by the transduced chimeric TCRs. The strong downregulation of the endogenous TCRs by the DN4.99 TCR is a characteristic that might improve the safety of adoptive cell therapy (ACT) in allogeneic HSCT, by reducing the

technology to prevent endogenous CD1 expression, prior to TCR transfer[6] (Supplementary Fig. 1d, e). Flow cytometry detection of the transduced chimeric TCRs by anti-mouse TCR Cβ monoclonal antibody (mAb) showed that four TCRs (DN4.99, PZP8A6, DN4.2, and P8E3) were efficiently expressed on the surface of the transduced Jurkat 76 β2m⁻ cells (Fig. 2a).

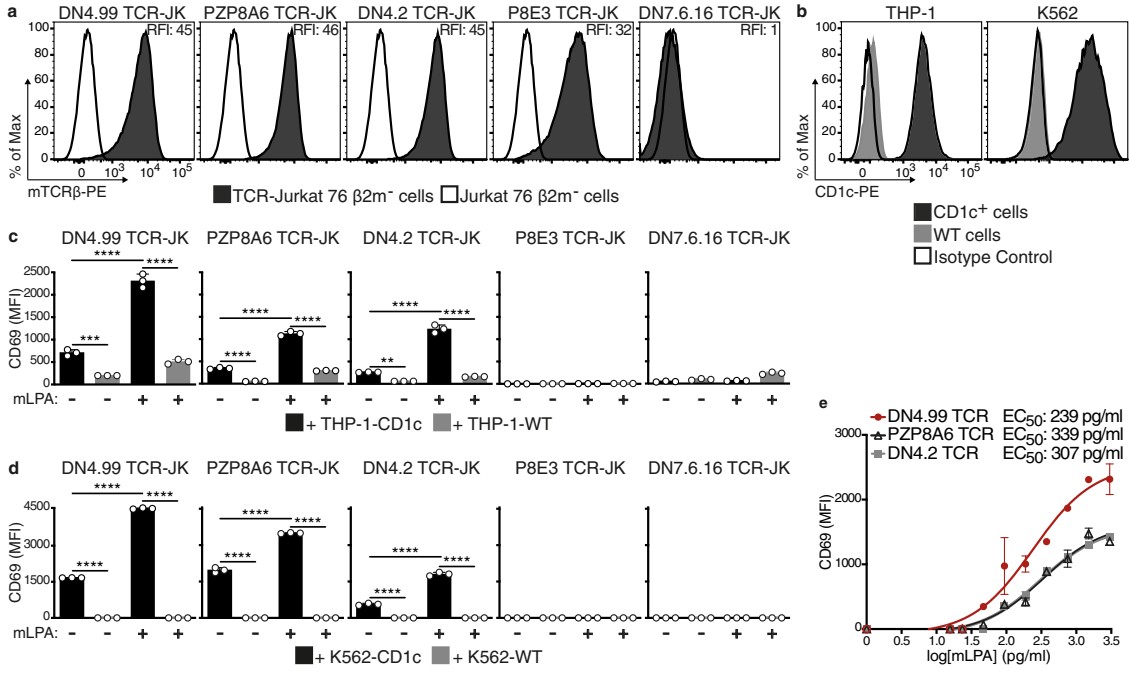

**Fig. 2 Identification of the lead CD1c-restricted mLPA-specific TCR to target acute leukemia.** Chimeric human V-mouse C TCR chains obtained from the indicated CD1c self-reactive T cell clones[10] followed by IRES and the GFP-coding sequence were cloned into lentivectors and transduced into Jurkat (JK) cells. Transduced JK cells were sorted to have comparable TCR expression. **a** TCR expression on transduced (black histograms) and non-transduced (white histograms) JK cells determined by flow cytometry labeling with anti-mouse TCRβ (mTCRβ) monoclonal antibody (mAb). Relative fluorescence Intensity (RFI) was calculated as the ratio between the intensity of labeling on transduced and non-transduced JK cells. **b** Flow cytometry detection of CD1c expression on CD1c+ (black histograms) and wt (gray histograms) THP-1 and K562 cells labeled with anti-human CD1c mAb. White histograms represent labeling with isotype-matched control mAb. **c, d** TCR-JK cells recognize CD1c+ leukemia cells. TCR-JK cells were cultured overnight with THP-1-CD1c (**c**) or K562-CD1c (**d**) leukemia cells ± 1.5 ng/ml of methyl-lysophosphatidic acid (mLPA, black bars), at Effector:Target (E:T) ratio of 2.5:1 (**c**) or 1:1 (**d**). Wild-type (WT) cells were used as a negative control (gray bars). JK cell activation was assessed as increased CD69 expression intensity as measured by flow cytometry (Mean Fluorescence Intensity, MFI, reported). Data are represented as mean ± SD. Results are representative of 2 (**c**) and 4 (**d**) independent experiments with $n = 3$ independent replicates per experiment. **P = 0.0023; ***P = 0.004; ****P < 0.0001 determined by Ordinary one-way ANOVA followed by Tukey's multiple comparison test. **e** mLPA recognition by the DN4.99 (red dots), PZP8A6 (black triangles) and DN4.2 (gray squares) TCR-JK cells, cultured overnight at 1:1 E:T ratio with K562-CD1c cells loaded with serial dilutions of mLPA. CD69 expression intensity on JK cells was then measured by flow cytometry. The interpolation curves of CD69 expression on TCR-JK cells in response to the indicated mLPA concentration are shown. $EC_{50}$ values were obtained as detailed in Methods. Data are represented as mean ± SEM. Results are representative of 3 independent experiments with $n = 3$ independent replicates per experiment.

expression of potentially alloreactive TCRs implicated in the pathogenesis of GvHD, although additional strategies to prevent GvHD by TCR-T cells are warranted for clinical application. The expanded TCR-transduced and non-transduced T cells reproducibly contained mixed central memory, effector memory, terminal effector, and CD95+ stem cell memory populations (Supplementary Fig. 2b, c), as expected by the action of IL-7 and IL-15 in culture[18]. We then tested the mLPA-specific TCR-transduced primary T cells (TCR-T cells) for their ability to recognize CD1c-expressing AML cell lines in vitro. DN4.99, PZP8A6, and DN4.2 TCR-T cells secreted IFN-γ in the presence of THP-1-CD1c, but not of THP-1 WT cells, in contrast to non-transduced T cells (Fig. 3c).

Taken together, these data identified DN4.99 as the lead TCR for further development due to its high level of expression, strongest endogenous TCR repression, strongest reactivity against CD1c-expressing leukemia cells and mLPA specificity.

**DN4.99 TCR-T cells recognize and kill human leukemia cells in vitro.** We next assessed the breadth of reactivity of DN4.99 TCR-T cells against a panel of cell lines representing AML (THP-1, MOLM-13, K562), ALL (NALM-6, CCRF-SB, MOLT-4), and B lymphoblastoid cells (C1R), which expressed at different levels natural or transduced CD1c (Fig. 4a). After 48 h of co-culture

with cell lines, DN4.99 TCR-T cells had secreted significantly more IFN-γ than non-transduced T cells, and this effect was dependent on CD1c (Fig. 4b). Similarly, in 1:1 effector:target (E:T) co-cultures, DN4.99 TCR-T cells had almost completely eliminated the target cell lines after 72 h, dependent on CD1c, while target cells remained abundant in co-cultures with non-transduced T cells (Fig. 4c–e). Because the DN4.99 TCR-T cell populations contained both CD4+ and CD8+ subsets, we assessed the relative cytokine production and killing ability of either subset by sorting and comparing them against THP-1-CD1c cells. We found that both T cell subsets were polyfunctional, with CD4+ cells producing a larger panel of Th1 and Th2 cytokines (Supplementary Fig. 3a–c). Both subsets eliminated the target cells but, as expected, CD8+ T cells were significantly more efficient at the lower E:T ratios of 1:5 and 1:10 (Supplementary Fig. 3d), also in eliminating MOLM-13 cells that express a lower CD1c level (Supplementary Fig. 3e).

Next, we asked whether DN4.99 TCR-T cells could recognize and kill primary CD1c+ leukemia blasts (Fig. 5a). Clinical evidence with CD19 chimeric antigen receptor (CAR)-T cell therapy in B cell malignancies, suggests that optimal efficacy was obtained by the transfer of a 4:1 mixture of CD8+ and CD4+ transduced T cells[19], owed to helper cytokine production by CD4+ T cells. Therefore, in light of the cytokine and

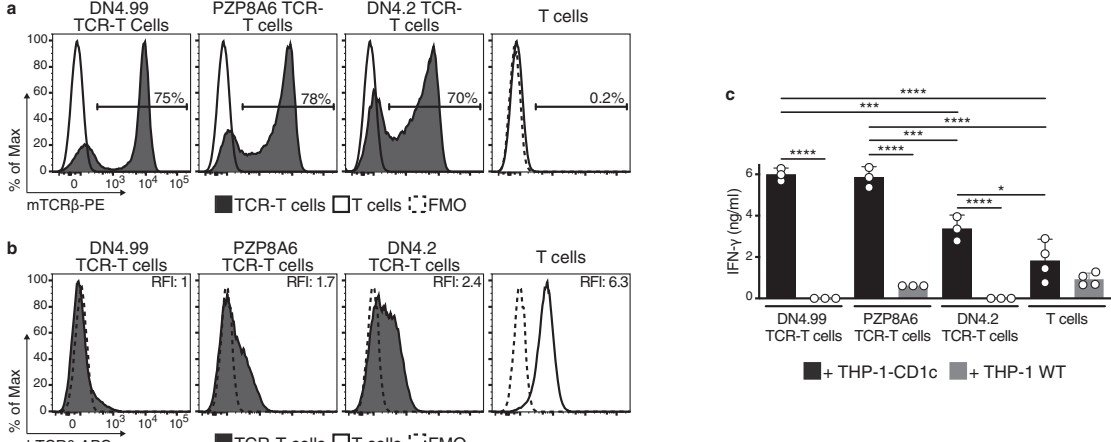

**Fig. 3 Primary T cell transduction confirm the DN4.99 TCR as lead CD1c-restricted mLPA-specific TCR to target acute leukemia.** DN4.99, PZP8A6, and DN4.2 TCRs were transduced into primary T cells. **a** Chimeric TCR expression on transduced (black histograms) and non-transduced primary polyclonal T cells (white histograms) determined by labeling with anti-mouse TCRβ (mTCRβ) monoclonal antibody (mAb). **b** Endogenous human TCR expression in chimeric TCR-transduced (black histograms) and non-transduced T cells (white histograms) determined at the same time point as in (**a**) by labeling with anti-human TCRβ (hTCRβ) mAb. Dotted histograms represent Fluorescence-Minus-One (FMO) labeling control. Relative Fluorescence Intensity (RFI) was calculated as the ratio between the intensity of labeling of the sample and control. **c** Recognition of THP-1-CD1c (black histograms) but not of THP-1-Wild-Type (WT) cells (gray histograms) by human primary T cells 18 days after their transduction with the indicated chimeric TCRs, determined by measuring the secreted IFN-γ in the supernatant of 2.5:1 Effector:Target ratio co-cultures. Data are represented as mean ± SD. Results are representative of 2 (for PZP8A6 TCR-T cell and DN4.2 TCR-T cells) or 4 independent experiments (for DN4.99 TCR-T cells) with $n = 3$ independent replicates per experiment. *$P = 0.0245$; ***$P = 0.0003$ DN4.99 TCR-T cells; ***$P = 0.0005$ PZP8A6 TCR-T cells; ****$P < 0.0001$ determined by Ordinary one-way ANOVA followed by Tukey's multiple comparison test.

killing results we obtained (Supplementary Fig. 3a–e), for these experiments we used a mixture of 70% CD8$^+$/30% CD4$^+$ DN4.99 TCR-T cells. Furthermore, we also increased the E:T ratio to 5–10:1, because primary leukemia blasts are more resistant to killing than cell lines. DN4.99 TCR-T cells specifically recognized all 9 blasts freshly isolated from the blood of patients, as indicated by CD1c-dependent IFN-γ secretion after 48 h of co-culture (Fig. 5b), and killed between 20 and 80% of the 6 primary leukemic blasts that could be assessed after 48 h of co-culture (Fig. 5c, d). Both recognition and killing were significantly reduced by the addition of anti-CD1c mAb. Thus DN4.99 TCR-T cells could recognize and kill all tested primary CD1c$^+$ AML and B-ALL blasts, further supporting the potential for ACT based on this TCR.

**DN4.99 TCR-T cells do not kill normal circulating CD1c$^+$ cells.** CD1c is expressed by normal circulating B cells, CD14$^+$ monocytes, and CD11c$^+$ DCs (a mixed cDC2/cDC3 and moDC population)[6,20], representing potential additional targets of DN4.99 TCR T-cells. To assess the risk of unwanted recognition of these cells, we isolated each population from eight independent healthy donors, loaded them or not with synthetic mLPA, and co-cultured them for 72 h with DN4.99 TCR-T cells at an E:T ratio of 1:1. Despite their expression of intermediate-high levels of CD1c (Fig. 6a–c), none of the cell populations were killed by DN4.99 TCR-T cells, unless monocytes and DCs were loaded with synthetic mLPA (Fig. 6d). This is consistent with our published data showing that these circulating leukocytes poorly stimulate CD1c self-reactive T cell clones because of their reduced mLPA content, in contrast to their malignant counterparts[10]. We saw similar results when a mixture of 70% CD8$^+$/30% CD4$^+$ DN4.99 TCR-T cells was used at an E:T ratio of 5:1 (Fig. 6e). DN4.99 TCR-T cells, therefore, spare healthy CD1c-expressing cells whilst effectively targeting CD1c-expressing leukemia blasts ex vivo, supporting the further preclinical investigation of cell therapy based on this TCR.

**ACT with DN4.99 TCR-T cells delays leukemia progression in mice.** Having demonstrated potent CD1c-expressing leukemia-cell specific cytotoxicity of DN4.99 TCR-T cells, we next assessed DN4.99 TCR-T cell in vivo antitumor activity in mice injected with human acute leukemia-cell lines (THP-1, NALM-6, MOLM-13) stably transduced to express CD1c. The cell lines were also engineered to express a secreted luciferase to enable non-invasive monitoring of tumor progression by measuring the ex vivo bioluminescence of serum samples[21]. Furthermore, we selected these cell lines to test the DN4.99 TCR-T cells against a diverse selection of pathological features and levels of aggression, representing the acute leukemia spectrum: following intravenous (i.v.) injection into NOD.Cg-Prkdcscid IL- 2rgtm1Wjl/SzJ (NSG) mice, the THP-1 AML cell line, which is the least aggressive, generates liver myeloid sarcomas, while the NALM-6 B-ALL and MOLM-13 AML cell lines home to the bone marrow and are of relatively moderate and high aggressiveness, respectively[22–24].

First, we injected $1 \times 10^6$ THP-1-CD1c cells i.v. into NSG mice that, after tumor engraftment confirmed by their serum luciferase levels, were randomized into three groups, and sub-lethally irradiated, followed 24 h later by the transfer i.v. of $1 \times 10^7$ DN4.99 TCR-T cells (70% CD8$^+$/30% CD4$^+$), the same number of T cells (70% CD8$^+$/30% CD4$^+$) from the same donor undergoing parallel activation and expansion but no TCR transduction, or vehicle only (Fig. 7a). DN4.99 TCR-T cell transfer significantly delayed leukemia progression (Fig. 7b) and conferred a significant advantage in overall animals' survival (Fig. 7c) compared to non-transduced T cell transfer. The livers of mice receiving DN4.99 TCR-T cells were significantly smaller than controls (Fig. 7d), due to reduced THP-1 myeloid sarcoma infiltration, and contained transferred T cells (Fig. 7e). This experiment showed that DN4.99 TCR-T cells could localize within human leukemia-bearing mouse livers, reduce tumor growth and prolong mouse survival. Notably, in the absence of high tumor burden, none of mice that received DN4.99 TCR-T cells showed weight loss, an effect of the xenogenic GvHD

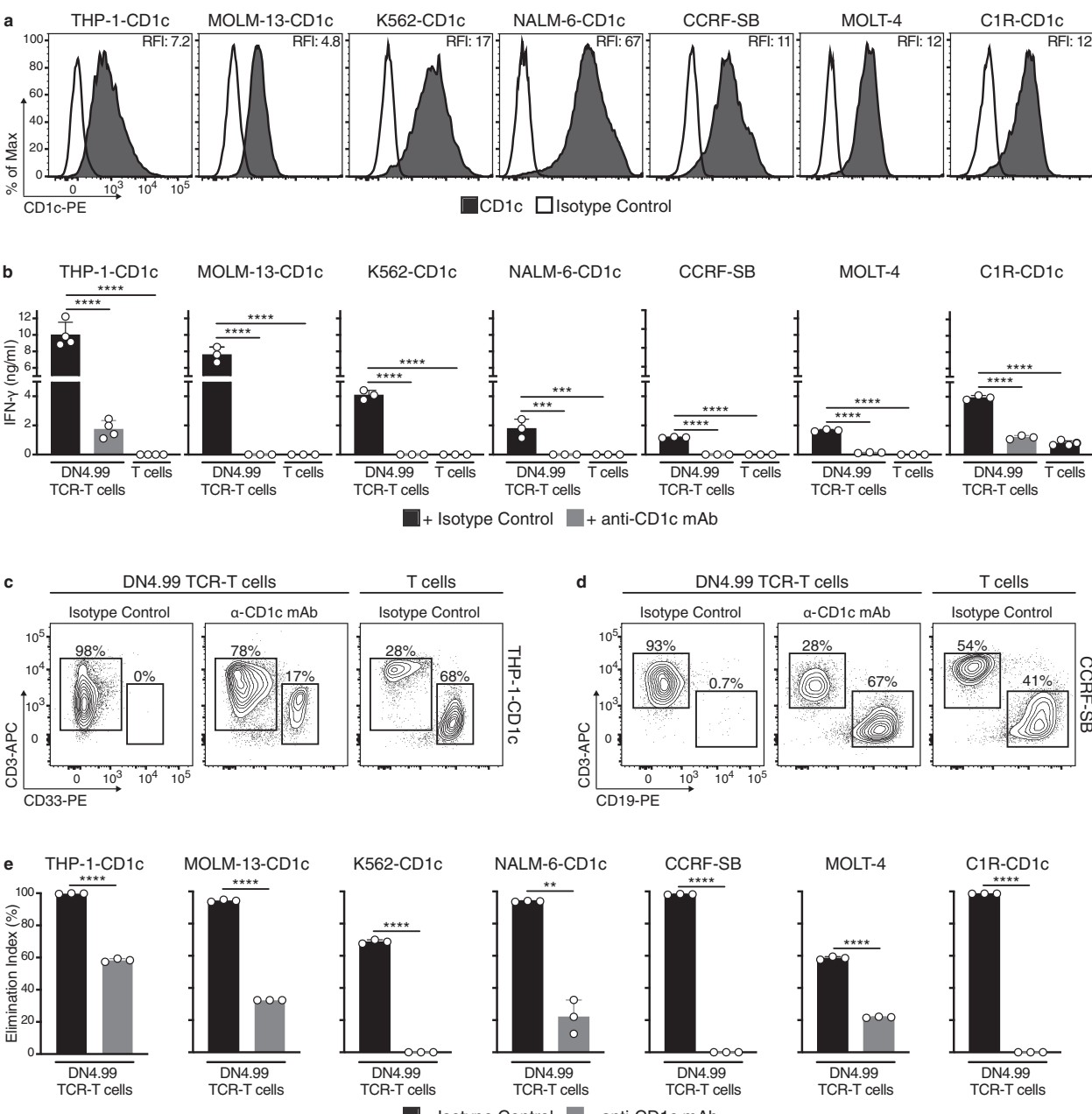

**Fig. 4 DN4.99 TCR-T cells efficiently recognize and kill CD1c⁺ leukemia-cell lines.** Primary T cells were transduced (DN4.99 TCR-T cells) or not-transduced (T cells) with the lead DN4.99 TCR and assessed in vitro for the recognition and killing of CD1c-expressing acute leukemia or lymphoblastoid cell lines. **a** CD1c expression (black histograms) on 7 different acute leukemia (THP-1, MOLM-13, K562, NALM-6, CCRF-SB, MOLT-4) or lymphoblastoid (C1R) cell lines, as determined by flow cytometry measurement of labeling with anti-human CD1c monoclonal antibody (mAb). White histograms represent labeling with isotype-matched control mAb. Relative Fluorescence Intensity (RFI) was calculated as the ratio between intensity of labeling of the sample and control. **b** Recognition of CD1c-expressing acute leukemia or lymphoblastoid cell lines by DN4.99 TCR-transduced or non-transduced primary T cells following 48 h of co-culture at a 1:1 Effector:Target ratio with either 20 µg/ml of blocking anti-CD1c mAb (gray bars) or an equivalent concentration of mouse IgG1κ isotype control (black bars). IFN-γ secretion into supernatants was measured by ELISA. Data are represented as mean ± SD. Results are representative of 2 (for CCRF-SB, MOLT-4 and C1R-CD1c) and 4 (for THP-1-CD1c, MOLM-13-CD1c, K562-CD1c and NALM-6-CD1c) independent experiments with n = 3 replicates per experiments. ***P = 0.0001; **** P < 0.0001 determined by Ordinary one-way ANOVA followed by Tukey's multiple comparison test. **c, d** Representative flow cytometry analysis to assess killing of THP-1-CD1c (**c**) and CCRF-SB (**d**) cells by DN4.99 TCR-T cells after 72 h of the same co-culture conditions of (**b**). **e** Killing of each target cell line in the same co-cultures determined as shown in (**c, d**), expressed as an Elimination Index as described in Methods. Gray bars represent co-cultures with anti-CD1c mAb, and black bars with the isotype control. Data are represented as mean ± SD. Results are representative of 2 (for K562-CD1c, CCRF-SB, MOLT-4 and C1R-CD1c) and 4 (for THP-1-CD1c, MOLM-13-CD1c and NALM-6-CD1c) independent experiments with n = 3 replicates per experiments. **P = 0.0071; ****P < 0.0001 determined by unpaired two-tailed Welch's t-test.

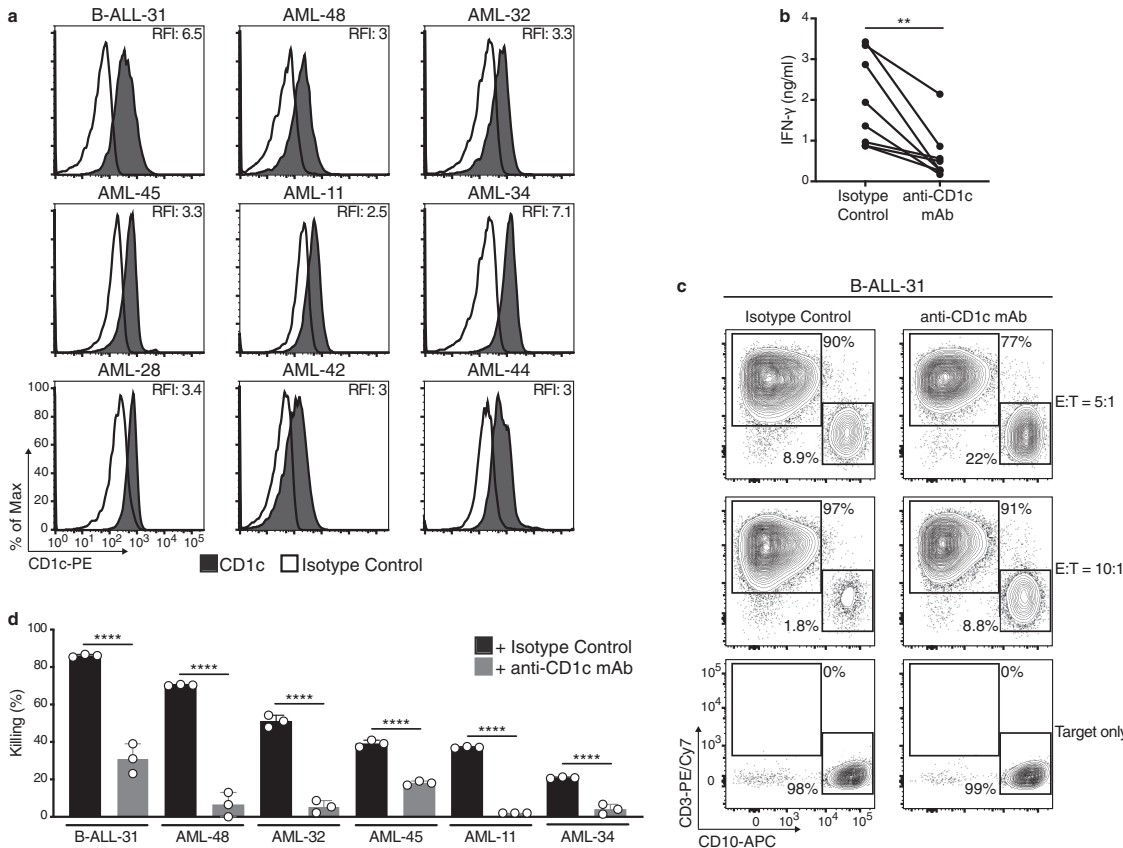

**Fig. 5 DN4.99 TCR-T cells efficiently recognize and kill primary CD1c⁺ Acute Leukemia Blasts in vitro.** Primary T cells were transduced with DN4.99 TCR (DN4.99 TCR-T cells), separated by cell sorting into CD4⁺ and CD8⁺ T cells, and mixed back again to generate a mixture of 70-80% CD8⁺ and 30-20% CD4⁺ cells, which was assessed in vitro for the recognition and killing of CD1c-expressing primary circulating acute leukemia blasts. **a** Flow cytometry detection of CD1c expression (black histograms) on ex vivo B-Acute Lymphoblastic Leukemia (B-ALL) and Acute Myeloid Leukemia (AML) blasts labeled with anti-human CD1c monoclonal antibody (mAb). White histograms represent labeling with isotype-matched control mAb. Relative Fluorescence Intensity (RFI) was calculated as the ratio between the intensity of labeling of sample and control. **b** Recognition of primary CD1c⁺ AML blasts by CD8⁺/CD4⁺ DN4.99 TCR-T cells, co-cultured for 48 h at 10:1 or 5:1 Effector:Target (E:T) ratio with either 20 µg/ml of blocking anti-CD1c mAb or an equivalent concentration of mouse IgG1κ isotype control. IFN-γ secretion into supernatants was measured by ELISA. **c** Representative killing of the target leukemia B-ALL-11 blasts by CD8⁺/CD4⁺ DN4.99 TCR-T cells, determined by flow cytometry labeling after 48 h of the same co-cultures. **d** Killing of each indicated primary blast by CD8⁺/CD4⁺ DN4.99 TCR-T cells of the same co-culture conditions of (**b**), expressed as % of killing, as described in Methods. Gray bars represent co-cultures with anti-CD1c mAb, and black bars with the isotype control. Allogenic CD8⁺/CD4⁺ DN4.99 TCR-T cells were from 3 independent healthy donors. Data are represented as mean ± SD. **P = 0.003 determined by unpaired two-tailed t-test followed by Mann–Whitney test (**b**). ****P < 0.0001 determined by Ordinary one-way ANOVA followed by Tukey's multiple comparison test (**d**).

commonly observed upon transfer of human peptide-specific TCR that cross-react with mouse peptide–MHC complexes[25], which was instead observed in mice receiving non-transduced T cells (Supplementary Fig. 3f).

We then assessed the ability of DN4.99 TCR-T cells to inhibit the growth of the more aggressive NALM-6 cell line in NSG mice. In this second model, we assessed the efficacy and safety of multiple transfers of DN4.99 TCR-T cells at 7 days intervals, to enhance the control of this relatively more aggressive leukemia (Fig. 8a). NALM-6-CD1c ($5 \times 10^5$) cells were injected i.v. into NSG mice that, after tumor engraftment confirmed by their serum luciferase levels, were randomized into three groups and sub-lethally irradiated, followed 24 h later by the transfer i.v. of $1 \times 10^7$ DN4.99 TCR-T cells; $1 \times 10^7$ T cells transduced with the chimeric TCR from the CD1c-restricted T cell clone DL15A31, specific for an M. tuberculosis-derived lipid and completely unable to react in vitro against CD1c-expressing leukemia cells ± mLPA (Supplementary Fig. 4 and ref. [10]); or vehicle alone (Fig. 8a). NALM-6-CD1c B-ALL cells engraft in the bone marrow of NSG mice, with the appearance of circulating blasts as the tumor burden increased[26]. Here, we saw

that leukemia progression was significantly delayed in mice that received DN4.99 TCR-T cells, compared to the two control groups (Fig. 8b). A second transfer of DN4.99 TCR-T cells, performed 7 days after the first one, further significantly delayed leukemia progression (Fig. 8b), as also shown by reduced frequency of circulating blasts at day +17 from tumor injection (Fig. 8c). DN4.99 TCR-T cells were found in the bone marrow four days after the first ACT (Fig. 8d), demonstrating tumor co-localization also in this model. Similarly, DN4.99 TCR-T cell treatment significantly prolonged the survival of tumor-bearing mice (Fig. 8e). At the experimental endpoint, the DN4.99 TCR-T cell transfer resulted in the survival of NALM-6 cells expressing lower levels of CD1c than tumors of control mice (Fig. 8f), confirming the CD1c-dependent mechanism of tumor recognition and the strong anti-leukemia effect of this TCR.

Finally, we investigated the possibility of boosting the tumor-specific response of DN4.99 TCR-T cells in leukemia-bearing mice with an mLPA therapeutic immunization between T cell treatments. We injected $5 \times 10^4$ highly aggressive MOLM-13-CD1c AML cells i.v. into NSG mice, then 17 days later subjected

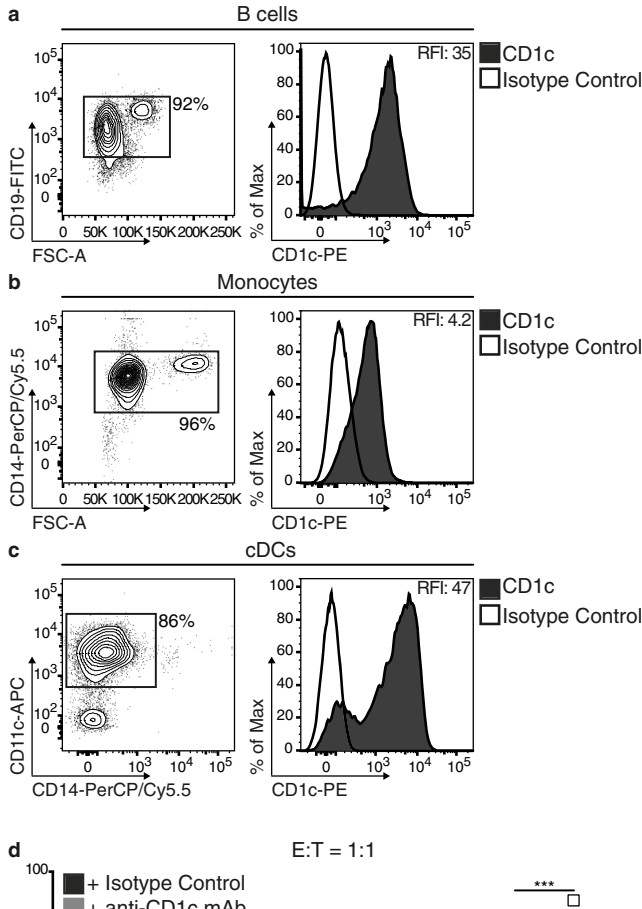

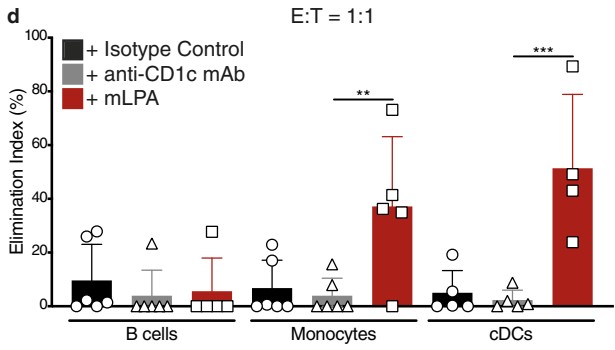

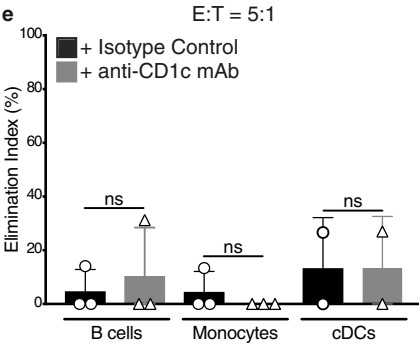

**Fig. 6 DN4.99 TCR-T cells do not recognize normal CD1c+ blood cells.** Primary DN4.99 TCR-transduced T (DN4.99 TCR-T) cells were assessed for killing in vitro normal circulating CD1c+ cells. **a–c** CD19+ B cells (**a**), CD14+ monocytes (**b**), and CD11c+ circulating dendritic cells (cDCs, **c**) were sequentially purified from the peripheral blood of healthy donors. Contour plots and histograms show the enrichment of each cell type and their CD1c expression (black histograms), determined ex vivo by flow cytometry labeling with the indicated monoclonal antibody (mAb). White histograms represent labeling with isotype-matched control mAb. Relative Fluorescence Intensity (RFI) was calculated as the ratio between the intensity of labeling of sample and control. **d** There was not any killing of primary B cells, monocytes and cDCs by DN4.99 TCR-T cells. T cells were co-cultured for 72 h with B cells, monocytes, or cDCs at a 1:1 Effector:Target (E:T) ratio, with either 20 µg/ml of blocking anti-CD1c mAb (gray bars), isotype control (black bars), or synthetic methyl-lysophosphatidic acid (mLPA; red bars), used as positive control. Target killing was determined by flow cytometry labeling and expressed as Elimination Index. **e** There was not any killing of primary B cells, monocytes and cDCs by DN4.99 TCR CD8+/CD4+ T cells. T cell co-cultures with B cells, monocytes, or cDCs and target killing assays were performed as described for primary leukemia blasts. Gray bars represent co-cultures with anti-CD1c mAb, and black bars with the isotype control. Note that DN4.99 TCR-T cells can only kill primary monocytes and cDCs upon supplementation with the synthetic mLPA-specific antigen. Data are represented as mean ± SD. **P = 0.0071; *** = 0.0001; not significant (ns) were determined by Ordinary one-way ANOVA followed by Tukey's multiple comparison test. Shown are results obtained in independent experiments with n = 3 replicates/each displayed as mean; (**d**) n = 4 experiments for cDCs + mLPA; 5 for B cells + mLPA, monocytes + mLPA, cDCs + Isotype control and cDCs + anti-CD1c mAb; and n = 6 for all the other conditions. **e** n = 2 for cDCs and n = 3 for B cells and monocytes. Normal circulating CD1c+ cells were purified from different healthy donors in each experiment.

are otherwise lacking in mice, as they do not have group 1 *CD1* genes. We found that the first adoptive T cell transfer (day 18 post-initiation) delayed leukemia progression up to day 24, after which progression of these highly aggressive tumors resumed (Fig. 9b inset). Immunization with mLPA-loaded moDCs (day 29) resulted in a deflection of the leukemia progression curve only in the mice that had received DN4.99 TCR-T cells (Fig. 9b), and was also associated with a significant increase in the frequency of TCR-T cells (Fig. 9c), which probably contributed to delayed tumor growth. The combination of multiple DN4.99 TCR-T cell injections and mLPA-loaded moDC immunization resulted in significantly increased survival compared to vehicle-only-treated controls (Fig. 9d).

Collectively, these results highlight the efficacy of immunotherapy combining ACT plus tumor antigen-specific vaccination and confirmed the strong potential of a donor-unrestricted ACT approach for acute leukemia with CD1c-retargeted T cells.

## Discussion

We have shown that targeting T cells against tumor-related lipid antigens presented by CD1c are a viable immunotherapy strategy for hematological malignancies, which are driven by the pathological counterpart of the cells that normally express CD1 group 1 molecules. This strategy is importantly different from adoptive immunotherapy with T cells engineered to express TCRs specific for MHC-presented peptides; rather, the approach we propose is comparable with CAR-T cell therapy because it has no histocompatibility barriers, owing to the lack of CD1c polymorphism[6]. Our strategy allows the generation of "universal" effector T cells for a donor-unrestricted application of ACT to any patient bearing CD1c-expressing leukemia: moreover, in an allogeneic

them to sub-lethal irradiation, followed 24 h later by randomization and i.v. injection of DN4.99 TCR-T cells or vehicle only. On day 29, $5 \times 10^5$ human CD1c+ monocyte-derived dentritic cells (moDCs) pre-loaded with mLPA were injected i.v. into all mice, followed by further TCR-T cells/vehicle only injections on days 36 and 42 post-initiation (Fig. 9a). These antigen-loaded moDC were intended to promote activation and expansion of DN4.99 TCR-T cells in vivo by acting as CD1c-expressing APCs, which

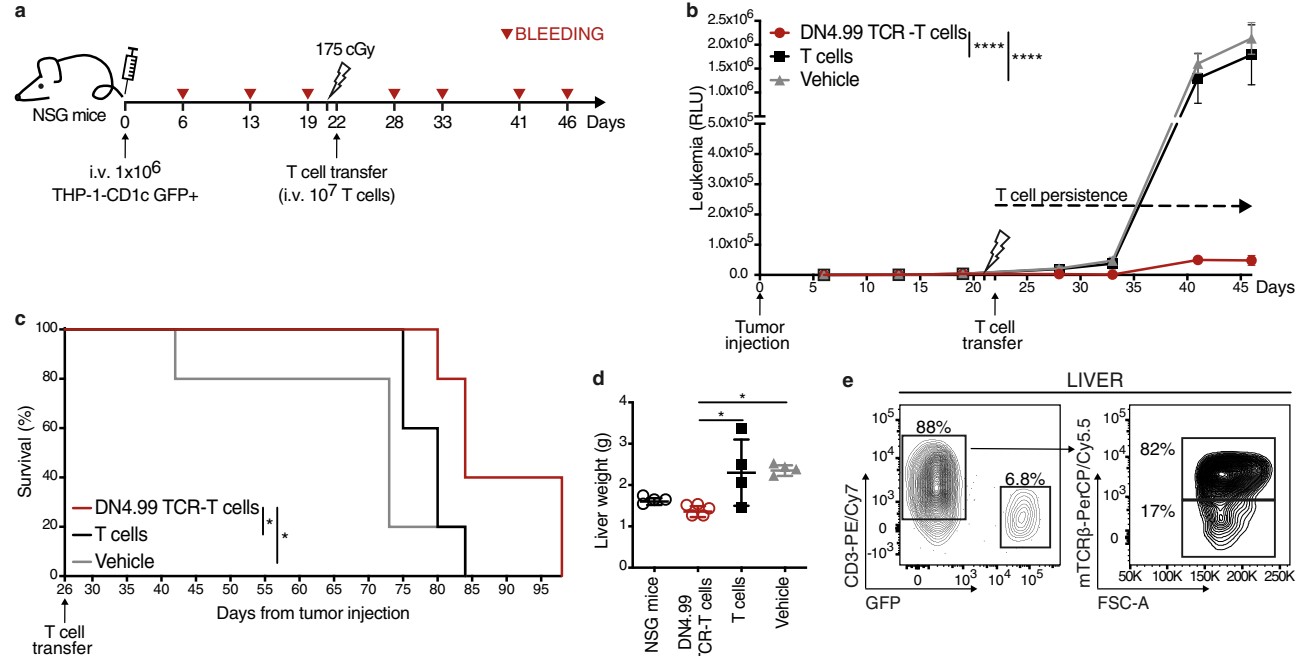

**Fig. 7 In vivo anti-leukemia efficacy of DN4.99 TCR-T cells against THP-1-CD1c-tumor cells.** DN4.99 TCR-transduced T (DN4.99 TCR-T) cells and non-transduced T cells (T cells), all at 70% CD8+/30% CD4+, were assessed for the ability to control the progression of THP-1-CD1c acute leukemia in vivo. **a** NOD.Cg-Prkdcscid IL- 2rgtm1Wjl/SzJ (NSG) mice (*n* = 15) received intravenous (i.v.) injection of 10^6 THP-1-CD1c cells co-expressing secreted LUCIA luciferase. Three weeks later the mice were sub-lethally irradiated, followed 24 h later by the transfer of 10^7 DN4.99 TCR-T cells (red lines/dots), T cells (black lines/squares), or vehicle (gray line/triangles). *n* = 5 mice/group. **b** Tumor progression was monitored weekly in blood collections by bioluminescence assay and depicted as the Relative Light Unit (RLU) detected at each time point. Data are represented as mean ± SEM. ****$P$ < 0.0001 determined on AUC by Ordinary one-way ANOVA followed by Tukey's multiple comparison test. **c** Kaplan–Meier survival curves show a significant increase in the survival of mice receiving DN4.99 TCR-T cells compared to the control groups. *$P$ = 0.0368 with T cells; *$P$ = 0.0274 with vehicle determined by log-rank (Mantel–Cox) test. **d** Reduced endpoint liver weight in mice receiving DN4.99 TCR-T cells (red dots) compared to the control groups (non-injected aged-match NSG mice: white dots; mice receiving: T cells, black squares; vehicle, gray triangles), determined at sacrifice (day +46). Data are represented as mean ± SD. *$P$ = 0.01557 with T cells; *$P$ = 0.0224 vehicle determined by Ordinary one-way ANOVA followed by Tukey's multiple comparison test. **e** Persistenc**e** of DN4.99 TCR-T cells in the liver of mice, determined by flow cytometry at sacrifice. DN4.99 TCR-T cells and THP1-CD1c leukemia cells were identified by mouse (m)TCRβ, human CD3, and GFP expression, respectively, among gated mouse CD45- mononuclear hepatic cells. Results are representative of 5 independent experiments with independent T cell lines giving comparable results.

HSCT setting, the risk of GvHD should be minimized upon ACT given the absence of CD1c on parenchymatous tissues.

Group 1 CD1-restricted T cells effectively extend the family of immune effectors recognizing non-polymorphic antigen-presenting molecules or ligands whose TCRs can be exploited to generate unrestricted antitumor effector cells. Our strategy provides additional treatment options to the current arsenal of TCR- and CAR-T cells, particularly for AML, where no CAR has been yet approved due to major remaining challenges, including the off-tumor expression of currently identified CAR targets (CD33, CD34, CD123, FLT3)[27], and is broadly applicable to either adult or pediatric B-ALL and AML patients owing to their common expression of CD1c molecules on blasts. In addition, any CD1c-expressing hematological malignancies, such as T-ALL and DLBCL, may also become possible therapeutic targets.

We provide preclinical data that show how primary human T cells from a range of healthy donors can be engineered to express a CD1c-restricted, mLPA-specific TCR and used for effective adoptive immunotherapy of acute leukemia in mice, without any clear side-effects assessable within the limitations of human xenograft models in immunodeficient animals. A first criterion in the development of such an approach is the ability to generate homogenous anti-leukemia T cell populations preferentially expressing the transduced recombinant TCRs; this aims at enhancing the therapeutic index of the ACT strategy, while reducing its toxicity by minimizing off-target recognition,

alloreactivity, and GvHD risks[28]. Here, we achieved this by generating chimeric molecules bearing the human TCR Vα and Vβ regions linked to the mouse C domains, which have two key advantages: they reduce the risk of potentially harmful mispairing between the endogenous and exogenous TCR chains[29–31]; and they increase the expression of the transduced TCR based on the preferential pairing of mouse C regions with the human CD3ζ chain, at the expense of the endogenous molecules[12,31]. The selected chimeric DN4.99 TCR exhibited the greatest ability to compete with the endogenous TCRs in transduced T cells, suggesting that it behaves as a "strong" TCR with intrinsic properties that favor its surface expression[31,32]. Notably, for clinical applications, mouse C regions do not seem particularly immunogenic when transferred into man: when 26 cancer patients were treated with autologous T cells engineered with a fully mouse tumor-specific TCR, none developed antibodies against the mouse C domain, and antibodies against the mouse V regions were detected in only 6 patients[33]. Nonetheless, additional strategies to prevent GvHD by the TCR-transduced T cells could be envisaged, such as knocking-down endogenous TCR expression using genome-editing approaches[34], or transferring the antitumor TCR into the unconventional iNKT, MR1- or γδ-T cell subsets that do not elicit GvHD upon allogeneic transfer because they are restricted to non-polymorphic molecules and express TCRs with strong intra-chain physical constraints, thereby minimizing mispairing with endogenous TCR chains[35,36].

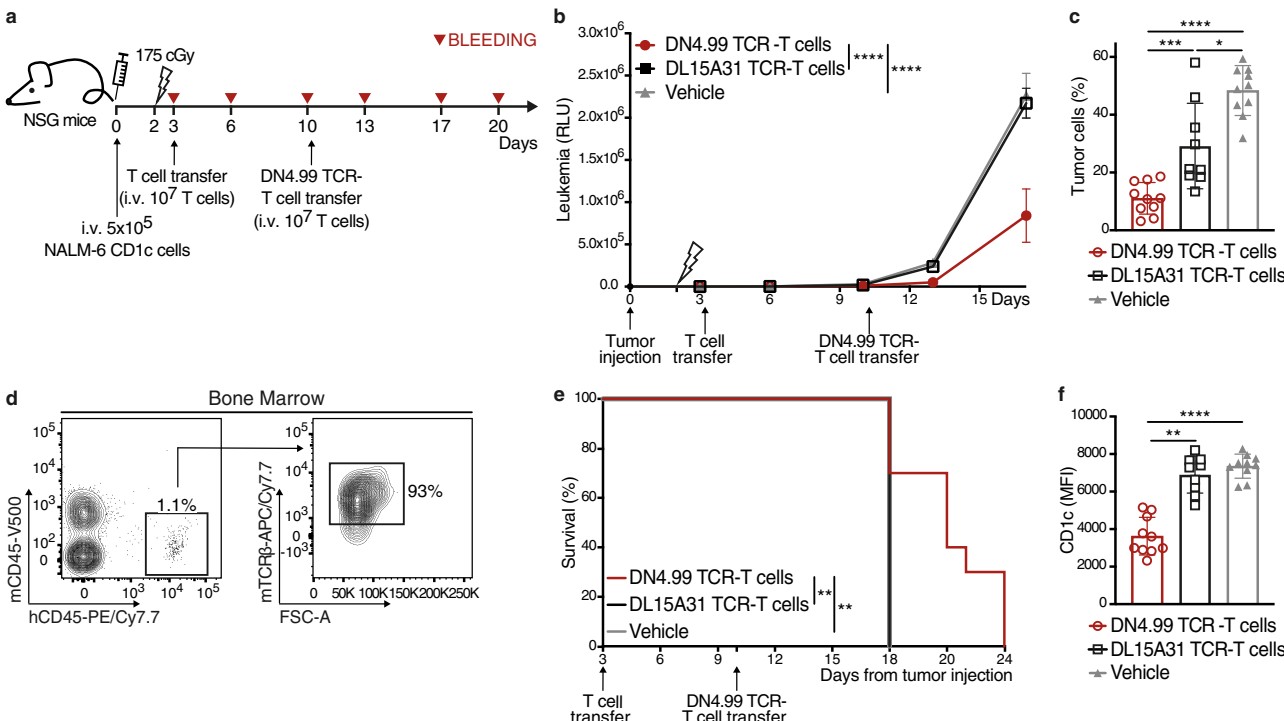

**Fig. 8 In vivo anti-leukemia efficacy of DN4.99 TCR-T cells against CD1c⁺ acute leukemia NALM-6.** DN4.99 TCR-transduced T (DN4.99 TCR-T) cells restrict NALM-6-CD1c leukemia progression in vivo. **a** NOD.Cg-Prkdcscid IL- 2rgtm1Wjl/SzJ (NSG) mice (n = 30) received intravenous (i.v.) injection of 5 × 10⁵ NALM-6 CD1c co-expressing secreted LUCIA luciferase. Two days later, the mice were sub-lethally irradiated, followed 24 h later by the transfer of 10⁷ primary 70% CD8⁺/30% CD4⁺ DN4.99 TCR-T cells (red lines/dots), control T cells (black line/squares) transduced with the CD1c-restricted, mycobacterial lipid-specific DL15A31 TCR (DL153A1 TCR-T cells; Supplementary Fig. 4 and ref. [10]) or vehicle (gray lines/triangles). n = 10 mice/group. After 7 days, DN4.99 TCR-T cells were transferred a second time, without prior irradiation, into the same mice. **b** Tumor progression was monitored in peripheral blood by a bioluminescence assay. RLU, relative light unit. Data are represented as mean ± SEM. ****P < 0.0001 determined on AUC by Ordinary one-way ANOVA followed by Tukey's multiple comparison test. **c** DN4.99 TCR-T cells reduce the number of circulating NALM-6-CD1c leukemia cells. The frequency of human CD19⁺ tumor cells was determined by flow cytometry in the blood of mice at day+17 from tumor injection. The percentage of the tumor cells was normalized based on the percentage of mouse CD45⁻ cells. *P = 0.0101; ***P = 0.0003; ****P < 0.0001. **d** Presence of DN4.99 TCR-T cells in the bone marrow of NALM-6 CD1c bearing mice, determined by flow cytometry 4 days after their adoptive transfer. DN4.99 TCR-T cells were identified as mouse (m)TCRb⁺ cells in the human (h)CD45⁺ mouse (m)CD45⁻ cells· **e** Kaplan-Meier survival curves show a significant increase in the survival of mice receiving DN4.99 TCR-T cells compared to control groups. **P = 0.0014 determined by log-rank (Mantel–Cox) test. **f** Reduced CD1c expression on NALM-6-CD1c cells that survive DN4.99 TCR-T cell, as determined by flow cytometry labeling (MFI, Mean Fluorescence Intensity) of human CD19⁺ NALM-6-CD1c cells in the circulation at day+17 from tumor injection. **P = 0.0015; ****P < 0.0001. Data in (**c**) and (**f**) are represented as mean ± SD and P-values were determined by unpaired two-tailed t test followed by Mann–Whitney test. Results are representative of 3 independent experiments with independent T cell lines giving comparable results.

The DN4.99 TCR-T cells specifically recognized and killed not only CD1c-expressing acute myeloid and lymphoblastic leukemia-cell lines, but also primary circulating AML and B-ALL blasts in vitro. However, the ACT strategy we envisage involves the use of a TCR which is autoreactive and therefore potentially harmful. Thus, a second criterion for its development is the strong specificity of the lead DN4.99 TCR for self-antigens highly enriched in cancer cells, compared to normal tissues. The recognition of CD1c⁺ leukemic cells by T cell engineered with the selected lead DN4.99 TCR was indeed increased by the addition of synthetic mLPA, which suggests strong leukemia antigen-specificity. Furthermore, one of the two CD1c self-reactive TCRs that did not recognize CD1c⁺ target cells, the P8E3 TCR, shared the same Vα and Vβ genes with the highly reactive DN4.99 TCR, hinting to a critical role of the CDR3 regions for the recognition of the mLPA–CD1c complex. This cumulative evidence strongly suggests that the selected TCR does not directly recognize CD1c, but depends rather on the association of mLPA with CD1c to react. Given that mLPA is highly enriched in malignant cells compared to normal[10], the requirement for mLPA presentation by CD1c adds another layer of protection and again favors the

proposed ACT approach. Consistent with the requirement for mLPA–CD1c complexes to stimulate the lead DN4.99 TCR, and with the need for the self-lipid antigen[10], normal circulating CD1c-expressing blood cells (B cells, monocytes, cDCs) were not killed by the transduced T cells, even at a high 5:1 E:T ratio, unless monocytes or DCs were artificially pre-loaded with mLPA. In normal conditions, therefore, the adoptively transferred DN4.99 TCR-T cells should spare normal cells, unlike CD19 CAR-T cell therapy which causes B cell aplasia[37], underscoring the likely safety of ACT targeting mLPA–CD1c complexes. Nevertheless, the safety of the engineered T cells could be further increased by introducing suicide or antibody-targeting genes into the TCR-expressing lentiviral coding vector, a strategy already exploited in TCR and CAR gene therapy[38–41].

The unique availability of a defined leukemia-enriched self-lipid antigen also provides us with the opportunity to boost the engineered T cell response with mLPA-containing vaccines, in turn stimulating and enhancing the efficacy of the CD1c-targeting ACT strategy. We used human moDCs as vehicles for synthetic mLPA as they can also provide high levels of CD1c expression and strong costimulatory signals. In mice, which lack group 1

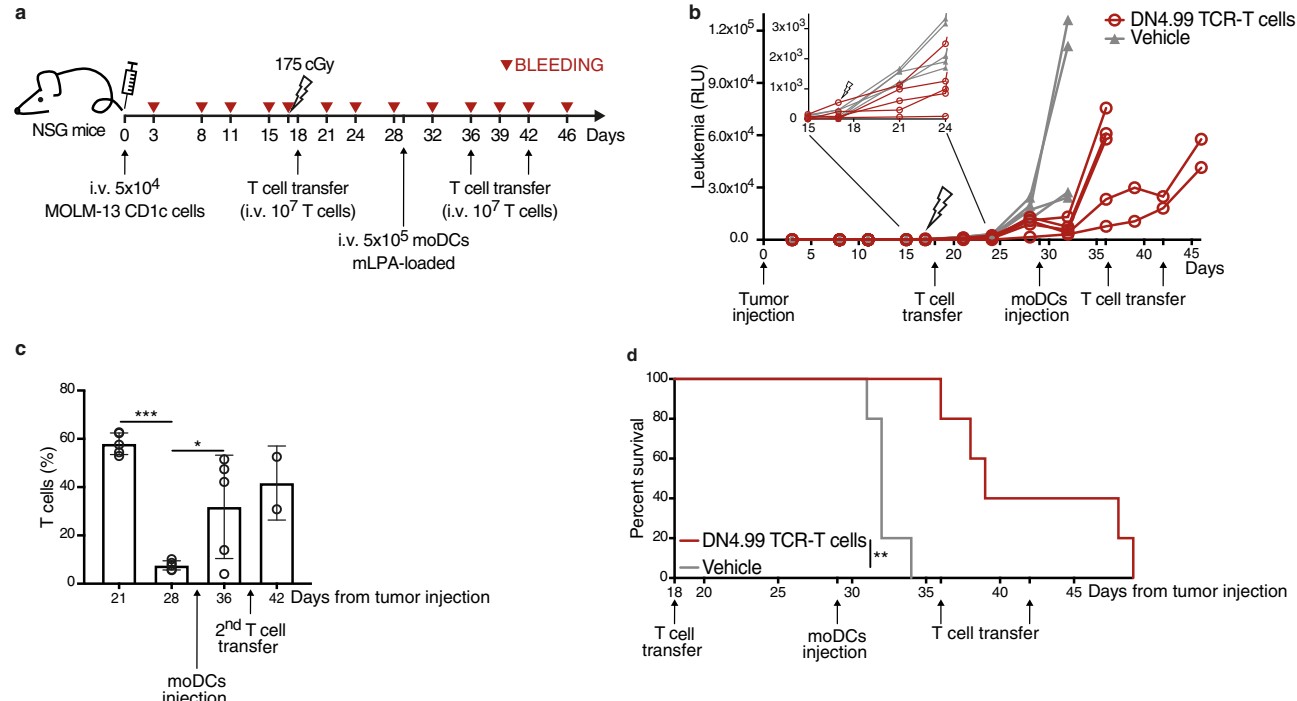

**Fig. 9 mLPA-antigen cellular immunization sustains the in vivo anti-leukemia efficacy of DN4.99 TCR-T cells against CD1c⁺ acute leukemia MOLM-13.**
**a** DN4.99 TCR-transduced T (DN4.99 TCR-T) cells restrict MOLM-13 leukemia progression in vivo. NOD.Cg-Prkdcscid IL- 2rgtm1Wjl/SzJ (NSG) mice (*n* = 10) received intravenous (i.v.) injection of 5 × 10⁴ MOLM-13-CD1c cells co-expressing secreted LUCIA luciferase. After 17 days, mice were sub-lethally irradiated, followed 24 h later by the transfer of 10⁷ DN4.99 TCR-T cells (red lines/dots) or vehicle (gray lines/triangles). n = 5 mice/group. After 2 weeks, all mice received i.v. 5 × 10⁵ monocyte-derived Dendritic Cells (moDCs) loaded with methyl-lysophosphatidic acid (mLPA), followed, only in the mice the had originally received the first T cell transfer, by two subsequent transfers of 10⁷ DN4.99 TCR-T cells one week apart. **b** Tumor progression was monitored in peripheral blood by a bioluminescence assay. RLU, Relative Light Unit. **c** DN4.99 TCR-T cell expansion upon i.v. injection of mLPA-loaded moDCs into MOLM-13-CD1c tumor-bearing mice, determined as the percentage of human T cells in the peripheral blood of mice by flow cytometry labeling with anti-mouse CD45 and anti-human CD3 monoclonal antibody. T cell frequency was normalized based on the percentage of mouse CD45⁻ cells. Data are represented as mean ± SD. *P = 0.0467; ***P = 0.0002 determined by Ordinary one-way ANOVA followed by Tukey's multiple comparison test. n = 5 mice at day +21, +26, and +37; n = 2 mice at day +42. **d** Kaplan–Meier survival curves show a significant increase in the survival of mice receiving DN4.99 TCR-T cells compared to control groups. **P = 0.002 determined by log-rank (Mantel–Cox) test. Results are representative of 2 independent experiments with independent T cell lines giving comparable results.

CD1 expression, this induced a marked increase in numbers of the transferred DN4.99 TCR-T cells and in their anti-leukemia functions. This vaccine strategy has already been approved for use in patients[42], and it is a suitable option to improve our ACT strategy. The presence of endogenous CD1c-expressing APCs in man may also support a cell-free formulation of an mLPA vaccine for its direct delivery in vivo. However, this approach should be carefully evaluated because it may associate with unwanted potential on-target/off-tumor elimination of the endogenous CD1c⁺ APCs presenting mLPA to engineered T cells.

A further advantageous feature of the DN4.99 TCR is its derivation from a CD4/CD8 double negative T cell clone, which enables its coreceptor-independent CD1c recognition and allows it to be used to efficiently redirect both CD8⁺ and CD4⁺ trans-duced T cells against CD1c⁺ leukemia targets. The following evidence from CAR-T cell therapy[19] and preclinical studies suggesting that CD4⁺ CAR-T cells, upon tumor recognition, produce IL-2 that augments CD8⁺ CAR-T cell proliferation, survival, and antitumor lytic activity[43], we also generated 70% CD8⁺/30% CD4⁺ mixture of DN4.99 TCR-T cells that was effective in vivo. Future studies could explore a range of relative ratios between the two subsets in order to understand the optimal point for antitumor efficacy.

Finally, our data establish the feasibility and preclinical safety of single, as well as repeated, administration of DN4.99 TCR-T

cells to control leukemia in vivo in three distinct cancer xenograft models. In all models, we saw that transferred T cells migrated to the tumor site, and all models showed clear clinical benefit from transfer of DN4.99 TCR-T cells. Interestingly, the tumor cells that survived in the mice had a lower expression of CD1c compared to remaining cells in the control groups, supporting the CD1c-dependent mechanism of tumor recognition and the strong anti-leukemia effect of this TCR, while also suggesting a possible mechanism of immune escape relying on the loss/reduction of the target of the immune response. Since the CD1c gene is identical on both human chromosomes 1[6], unlike most HLA alleles on chromosome 6, we hypothesize that the loss of one allele might cause a reduction in CD1c expression but not its complete loss on the leukemia surface, preserving its recognition by DN4.99 TCR-T cells. It is also possible that the physiological expression of CD1c in blasts might be epigenetically or transcriptionally regu-lated. It will be critical to unravel these largely unknown mechanisms in order to define possible pharmacological inter-ventions to sustain CD1c expression specifically in blasts. Nevertheless, the immune evasion phenomenon is frequently observed in other current ACTs that show clinical efficacy, such as the B-ALL escape from CD19 CAR-T cells via CD19 downregulation[44], or the HLA loss that occurs in acute leuke-mia relapse resulting from allogeneic T cell immune pressure after allogeneic hematopoietic cell transplantation[45]. Further

studies are warranted to improve the DN4.99 TCR affinity for the mLPA–CD1c complex, or to pharmacologically sustain CD1c expression on malignant cells, in order to target cancer cells expressing low CD1c levels and prevent/minimize possible immune evasion. Furthermore, co-targeting CD1c with other already available leukemia-restricted antigens could also minimize the antigen loss leading to immune evasion.

In conclusion, our results support the use of mLPA-specific CD1c-restricted T cell as an attractive option for adoptive immunotherapy of leukemia across MHC barriers.

## Methods

**Animal care**. All mouse experiments performed in this study were approved by the San Raffaele Scientific Institute IACUC (678 and 1072) and by the Italian Ministry of Health (Rome, Italy) and were conducted in compliance with national laws and policies. Mice were housed in a Specific and Opportunistic Pathogen Free (SOPF) with optimize and controlled conditions (12-h light/12-h dark cycle, temperature of 22 °C ± 2 °C, and relative humidity of 55% ± 5%), and monitored daily for health status and weight loss. Blood was collected once (for the THP-1 model) or twice (for NALM-6 and MOLM-13 models) per week to assess tumor progression. The terminal disease stage was determined when the luciferase signal in the blood reached $2$–$3 \times 10^6$ relative light unit (RLU) for THP-1 and NALM-6, and $1$–$2 \times 10^5$ RLU for MOLM-13. Mice were euthanized before study termination if they showed >20% body weight loss from their starting weight, or other distress signs (scruffy coat, altered postured, or reduced mobility).

**Human samples**. Peripheral blood containing primary leukemia blasts was obtained from adult patients (AML-48, AML-32, AML-45, AML-34, AML-28, AML-42, AML-44, San Raffaele Leukemia Biobank) following written informed consent, and from pediatric patients (B-ALL-31, Fondazione IRCCS Policlinico San Matteo Pavia, Italy; AML-11 M. Tettamanti Research Center, University of Milano-Bicocca, Monza, Italy) with written informed consent from their parents, and in agreement with the declaration of Helsinki. The study protocol "CD1TARGET" was approved by the San Raffaele Ethics Committee on November 6th, 2018. Primary leukemia samples were obtained from peripheral blood of male and female patients at diagnosis. The age range of adult patients was between 25 and 88 years, while that of pediatric patients was 3–10 years. Patients were recruited at diagnosis, with a percentage of circulating blasts between 60 and 97% of leukocytes to minimize manipulation following density gradient centrifugation. Moreover, only patients with CD1c expression on blasts with Relative fluorescence intensity ≥ 2.5 were recruited.

T cells, B cells, monocytes, and cDCs were purified from PBMCs isolated from buffy coats of anonymous healthy volunteers obtained after written informed consent.

**In silico measurement of CD1 gene expression in leukemia datasets**. The analyses of gene expression data from publicly available human leukemia datasets were performed in R (version 3.4.4) using Bioconductor libraries of BioC 3.6 and R statistical packages. Microarray raw data were download as CEL files from Gene Expression Omnibus using the GEOquery (v. 2.46) package[46]. The raw intensity signals were extracted from CEL files and normalized using the robust multi-array average procedure RMA[47] of the *affy* (v. 1.56) package. Specifically, probe fluorescence intensities were background adjusted, normalized using quantile normalization, and log2 expression values calculated using median polish summarization and the custom chip definition files from Brain Array (v. 20.0, http://brainarray.mbni.med.umich.edu/Brainarray/Database/CustomCDF) for human Affymetrix arrays based on Entrez genes (i.de., hgu133ahsentrezgcdf, hgu133bhsentrezgcdf, hgu133plus2hsentrez and hugene10sthsentrezg). When multiple array versions were used in a single dataset, e.g. GSE12417, only common probes across platforms were retained and batch correction was performed using the ComBat function of the SVA (v. 3.26.0) package[48] with the platform version as batch covariate. Raw data for the TCGA-LAML dataset were downloaded as raw counts from the TCGA repository using the TCGAbiolinks R package (v. 2.15)[49]. Data normalization was performed using the edgeR R package (v. 3.20.0;[50]). Specifically, raw counts were normalized to counts per million mapped reads (CPM) and only genes with a CPM greater than 1 in at least 3 samples were retained. Differential analysis of CD1c expression between each TCGA tumor and its corresponding normal tissue from GTEx was performed using the Gene Expression Profiling Interactive Analysis web tool (GEPIA v.1.0; http://gepia.cancer-pku.cn;[51]). In GEPIA, gene expression levels of samples from the two projects were re-computed using a uniform pipeline. CD1c expression was considered significant at $P \le 0.01$ in a one-way ANOVA.

**Lentivirus construction and production**. The sequences encoding the chimeric TCRs were synthetized by GeneArt. A Kozak consensus sequence (accgcc) was inserted 5′ to the ATG to optimize the translational starting site, and a STOP codon was inserted at the 3′ of the Cβ. The cDNAs of the α and β TCRs were linked by the 2A peptide and subcloned into the LV pHRSIN-Bx-IRES-EmGFP (kindly provided

by Dr. V. Cerundolo, University of Oxford), in which the TCR-coding genes can be followed by IRES and the GFP-coding sequence. The cDNA encoding CD1c was cloned by PCR with specific primers (listed in Supplementary Table 3) and inserted into the LV pHRSIN-Bx-IRES-EmGFP. The production of the lentiviral vectors was performed as described previously[52] by the transient transfection of 293T cell line (ATCC) with a second-generation lentiviral vector system. Subconfluent 293T, seeded in a 15 cm dish in IMDM medium (Lonza) supplemented with 10% heat-inactivated FBS (EuroClone), 10U/ml penicillin and streptomycin (Lonza), were transfected by $Ca_2PO_4$ method with the packaging plasmids (12 μg pMD2.VSV-G, 16.25 μg pCMVΔR8.74, 6.25 μg pRSV-rev) and 32 μg of transfer vector plasmid. After 16 h medium was replaced and 30 h later the supernatant containing the LV were collected and concentrate by ultracentrifugation at 20,000 rpm for 2 h.

**Leukemia-cell lines**. Cell lines were cultured in RPMI 1640-GlutaMAX (Gibco) complete medium (supplemented with 10% heat-inactivated FBS (EuroClone), 10U/ml penicillin and streptomycin (Lonza), 1% non-essential amino acids 100x (Gibco), 1 mM sodium-pyruvate (Gibco), and 50 μM 2-mercaptoethanol (Gibco). Disruption of *B2M* gene in Jurkat 76 cells (kindly provided by Prof. H.J. Stauss, University College London) was performed by CRISPR/Cas9 technology with Dr. A. Lombardo (IRCCS San Raffaele Scientific Institute). Jurkat 76 β2m- cells were transduced with LV encoding the CD1c-restricted TCRs and GFP at a multiplicity of infection (MOI) of 100. The following leukemia-cell lines were used: THP-1-WT, K562-WT, MOLT-4, (American Type Culture Collection, Manassas, VA), K562-CD1c[8] (provided by Prof. D.B. Moody, Brigham and Women's Hospital, Harvard Medical School, Boston), C1R-CD1c (provided by Prof. S. Porcelli, Albert Einstein College of Medicine, New York) and MOLM-13 (provided by Dr. R. Bernardi, San Raffaele Scientific Institute), THP-1-CD1c, CCRF-SB, NALM-6 and THP-1 expressing the Gaussia LUCIA. MOLM-13 cells were transduced with a LV encoding the secreted Gaussia Luciferase Lucia and the LNGFR selection marker. Gaussia Luciferase Lucia-expressing THP-1, NALM-6, and MOLM-13 cells were transduced with a LV encoding CD1c (MOI:100). All cells were maintained at 37 °C in a humidified atmosphere containing 5% $CO_2$.

**TCR-transduced Jurkat 76 β2m- cell assays**. $10^5$ TCR-JK cells were co-cultured overnight at different E:T ratios with the indicated leukemia-cell lines (THP-1, 2.5:1 E:T ratio; or K562, 1:1 E:T ratio) that had been pre-loaded or not with different concentrations of mLPA for 4 hours at 37 °C. TCR-JK activation was measured by CD69 expression level using flow cytometry (expressed as median fluorescence intensity, MFI). The mLPA $EC_{50}$ for each transduced TCRs (Fig. 2e) were calculated using PRISM_V8 software (GraphPad), which plot the MFI of CD69 against the mLPA concentration in log scale, generating a sigmoidal curve.

**Primary T cell Culture, Transduction and Stimulation**. Peripheral blood mononuclear cells were isolated by Ficoll-Paque (GE HealthCare) density gradient centrifugation of buffy coats from healthy volunteers. $CD8^+$ and $CD4^+$ T cells were isolated using CD4 Isolation Kit (Miltenyi Biotec, Cat# 130-096-533) as shown in Supplementary Fig. 3a. T cells were activated with Dynabeads human T-Activator CD3/CD28 (Thermofisher, Cat# 11131D) at a 3:1 bead:cell ratio for 2 days, then transduced with the LV encoding the indicated TCR (MOI:100) and further expanded until day 18-20 before use (or cryopreservation). When indicated $CD8^+$ and $CD4^+$ T cells were combined to have 70% CD8 $^+$/30% CD4$^+$ DN4.99 TCR-T cells. T cells were activated and cultured in RPMI 1640-GlutaMAX medium supplemented with 10% heat-inactivated FBS, 10U/ml penicillin and streptomycin, 1% non-essential amino acids 100X, 1 mM sodium-pyruvate, 50 μM 2-mercaptoethanol, and 5 ng/ml rhIL-7 + rhIL-15 (R&D), approximately $22 \times 10^2$ IU/ml each.

**TCR-transduced T cell assays**. TCR-transduced and non-transduced T cells were co-cultured with leukemia-cell lines, primary leukemia blasts, or primary circulating B cells, monocytes, or cDCs at the indicated E:T ratios. Normal $CD1c^+$ APCs were subsequently purified from peripheral blood of healthy volunteers using immunomagnetic beads according to the manufacturer's protocols: first we purified monocytes with anti-CD14 microbeads (Miltenyi Biotec, Cat# 130-050-201); then we purified cDCs with anti-CD11c-APC mAb (diluted 1:100; Biolegend, clone 3.9) and anti-APC microbeads (Miltenyi Biotec, Cat# 130-090-855); finally, we purified B cells with anti-CD19 microbeads (Miltenyi Biotec, Cat# 130-050-301). Cultures also included either 20 μg/ml of anti-CD1c antibody (clone M241; Santa Cruz Biotechnology Inc.) or equivalent concentration of mouse IgG1κ isotype control (clone 107.3; BD). Where indicated, T cells were combined to obtain 70-80% $CD8^+$ and 20–30% $CD4^+$ cell mixtures. Supernatants were collected after 48 h to quantify IFN-γ release by enzyme-linked immunosorbent assay (ELISA). ELISA plates (Maxisorp, NUNC) were coated with mouse anti-human IFN-γ mAb (Invitrogen, diluted 1:1000, clone 2G1). After blocking with PBS1x-5% BSA, samples were added for 1 h and subsequently incubated with the mouse anti-human IFN-γ biotin-labeled mAb (Invitrogen, diluted 1:1000, clone B133.5). Signal was revealed by streptavidin-bound horseradish peroxidase (diluted 1:30000, Thermo Scientific) and TMB substrate (Sigma). The colorimetric reaction was stopped by the addition of 18% sulfuric acid and the absorbance at 450 nm optical density was read with

the iMark Microplate Reader (BIO-RAD, Microplate Manager Software v6.3) at 450 nm optical density and cytokine concentration was calculated based on the standard curve.

To assess killing, after 48 h (leukemia blast targets) or 72 h (leukemia-cell line targets) of co-culture cells were counted, and T and target cells were discriminated by flow cytometry analysis with cell-type-specific markers. The elimination index (EI) was calculated as 1−(number of residual target cells in the presence of transduced T cells/number of residual target cells in the presence of non-transduced T cells)*100. Percentage of killing was calculated as 1−(number of residual target cells in presence of transduced T cells/number of residual target cells alone)*100.

**Flow cytometry analysis**. CD1c expression was detected with anti-CD1c PE mAb from Santa Cruz Biotechnology Inc. (clone L161, diluted 1:20), or with anti-CD1c PE from BioLegend (clone L161, diluted 1:50) for the B-ALL-31. The isotype-matched control mAb used for this labeling was the PE mouse IgG1, κ isotype control (clone MOPC-21, BD, diluted 1:20).

Human leukemia cells were identified using the following anti-human mAbs (all from BioLegend unless otherwise stated): CD45 PE/Cy7 (clone HI30, diluted 1:200); CD33 FITC (clone HIM3-4, diluted 1:100), PE or APC (clone WM53, diluted 1:100); CD34 FITC or APC (clone 581, diluted 1:100); CD19 FITC, PE, PerCP/Cy5.5 or APC/Cy7 (clone HIB19, diluted 1:100); CD10 APC (clone HI10a, diluted 1:100), and CD1d-PE (clone 42.1, BD, diluted 1:20).

T cells were labeled with the following mAbs (all from BioLegend unless otherwise stated): anti-mouse TCRβ chain PE, PerCP/Cy5.5 or APC/Cy7 (clone H57-597, diluted 1:100); and anti-human TCRα/β-APC (clone IP26, diluted 1:20); CD3 FITC, PerCP/Cy5.5, PE/Cy7, APC or APC/Cy7 (clone HIT3a, diluted 1:100); CD4 FITC, PE/Cy7, APC (clone RPA-T4, diluted 1:100) or V500 (clone RPA-T4, BD, diluted 1:100); CD8 PE, PE-Cy7, APC or APC/Cy7 (clone SK1, diluted 1:100); CD69 APC (clone FN50, diluted 1:100); HLA-ABC APC (clone W6/32, diluted 1:100); β2m PE (clone 2M2, diluted 1:50); CD62L PE/Cy7 (clone DREG-56, diluted 1:100); CD54RA FITC (clone HI100, diluted 1:50); CCR7 PE (clone G043H7, diluted 1:50); CD95 PE (clone DX2, diluted 1:100).

Mouse cells were detected with anti-mouse CD45 APC/Cy7 (clone 30-F11, BioLegend, diluted 1:100) or V500 (clone 30-F1, BD, diluted 1:50).

Circulating human B cells, monocytes, and DCs were identified using the following mAbs (all from BioLegend): anti-human CD19 FITC, PE or PerCP/Cy5.5 (clone HIB19, diluted 1:100), CD20-PE/Cy7 (clone 2H7, diluted 1:100), CD14-PerCP/Cy5.5 (clone M5E2, diluted 1:100), and CD11c-APC (clone 3.9, diluted 1:100). Viable and dead cells were discriminated by DAPI staining. Samples were acquired on a FACS Canto II cell analyzer (BD, FACSDiva v8.0.2) and all data were analyzed with FlowJo_V10 software (BD). All gating strategies are shown in Supplementary Figs. 5 and 6. RFI (relative fluorescence intensity) was calculated as the ratio between the MFI of sample and control labeling (isotype-matched control mAb or fluorescence-minus-one (FMO) labeling control), unless otherwise stated.

**Human leukemia xenograft models**. CD1c and Lucia luciferase-expressing THP-1 (1 × 10^6/mouse), NALM-6 (5 × 10^5/mouse), or MOLM-13 (5 × 10^4/mouse) cells[22–24] were injected intravenously into 8-week-old male NOD.Cg-Prkdcscid IL-2rgtm1Wjl/SzJ (NSG) mice (Charles River). Tumor progression was monitored once (THP-1) or twice a week (NALM and MOLM-13) by assessing serum secreted Lucia luciferase bioluminescence, quantified with the QUANTI-Luc detection reagent (InvivoGen, rep-qlc1) using a luminometer Mithras (BertholdTech, MikroWin 2000 v4.41), and expressed as Relative Light Unit (RLU), according to the manufacturer's instructions. At tumor engraftment, confirmed by the presence of detectable serum luciferase level, mice were sub-lethally irradiated (175 cGy), and 24 h later were randomized to receive an intravenous injection of 1 × 10^7 70% CD8+/30% CD4+ TCR-transduced T cells (DN4.99 TCR-T or DL15A31 TCR-T), non-transduced T cells (T cells), or vehicle alone (PBS1x). T and leukemia-cell expansion in mouse blood was monitored weekly by flow cytometry. MoDCs were differentiated in vitro from purified CD14+ monocytes, that were cultured in RPMI 1640-GlutaMAX medium supplemented with 10% heat-inactivated FBS, 10U/ml penicillin in the presence of 50 ng/ml GM-CSF and 20 ng/ml IL-4 (Peprotech) for 6 days as described[53].

Mice were sacrificed when tumor growth reached the threshold value of 2–3 × 10^6 RLU for THP-1 and NALM-6, 1–2 × 10^5 RLU for MOLM-13, or when manifesting signs of suffering. The area under the curve (AUC) describing the kinetics of tumor growth was calculated for each mouse, and values derived from each group of animals were statistically compared.

**Statistics**. All statistical analyses were performed with Prism_V8 Software (GraphPad). Data are shown as Mean ± SD or shown as ± SEM with at least $n = 3$ replicates as indicated in the figure legends. All statistical tests used are indicated in the figure legend corresponding to each specific experiments. Comparisons were made using the unpaired two-tailed t-test between two group comparison or the one-way ANOVA test followed by Tukey's multiple for three or more groups. Animal survival data were analyzed using log-rank (Mantel–Cox). Differences with a $P$ value < 0.05 were considered statistically significant.

**Reporting summary**. Further information on research design is available in the Nature Research Reporting Summary linked to this article.

## Data availability
All data supporting the findings of this study are available within the paper and its supplementary information files or from the corresponding authors upon reasonable request. Microarray raw gene expression data (.CEL files) of all series listed in Supplementary Table 1 are available in Gene Expression Omnibus (https://www.ncbi.nlm.nih.gov/geo/). Raw data of the TCGA datasets are freely accessible at TCGA (https://www.cancer.gov/about-nci/organization/ccg/research/structural-genomics/tcga) and they have been download from the repository using the TCGAbiolinks R package. The primary accession codes and hyperlinks of GEO and TCGA datasets referred to in this manuscript are listed in Supplementary Table 1. Source data for all figures are provided within the paper, except for Fig. 1c since the publicly available online tool utilized to generate the graphs (http://gepia.cancer-pku.cn, as indicated in methods) does not make them available. Source data are provided with this paper.

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

## Acknowledgements

The study was funded by grants from Associazione Italiana Ricerca sul Cancro (AIRC) IG2017-ID.20081, Leukemia Lymphoma Society TRP 6481-16, Worldwide Cancer Research 19-0133 to G.C.; and by an AIRC-FIRC Fellowship to M. Consonni (number 16537). The authors gratefully acknowledge Dr. Angelo Lombardo and Dr. Angelo Amabile (IRCCS San Raffaele Scientific Institute) for the help in the disruption of *B2M* gene in Jurkat 76 by CRISPR/Cas9 technology. We thank Dr Lucy Robinson of Insight Editing London for editing of the manuscript prior to submission. Dr. Michela Consonni performed experiments of the study as fulfillment of her PhD degree of the International PhD School of Molecular Medicine, Università Vita-Salute San Raffaele, Milan, Italy.

## Author contributions

M. Consonni., P.D., G.C. supervised the study, analyzed the data and wrote the manuscript. M. Consonni., C.G., C.dL., A.M performed the experiments. A.G., S.B. performed computational analysis. D.M., M.S., F.C., M. B., S.M. took care of clinical management and sample collection. M. Casucci, D.H., G.DL., L.M., and C.B. provided critical reagents. All authors discussed data. D.M, M.S., C.dL., A.G., S.B., S.M., G.DL., L.M. contributed to manuscript revision. P.D., G.C. conceived the study and provided the funding.

## Competing interests

P.D, G.C., C.G., C.dL., M. Consonni are inventors on an international patent application PCT/EP2016/073584 submitted by San Raffaele Scientific Institute. All other authors declare no competing interests.
