## [Peer Review File · Nature Communications]

REVIEWER COMMENTS

Reviewer #1 (Remarks to the Author):

In the study by Consonni and colleagues, the CD1c mediated anti-tumour immune response is investigated for its therapeutic potential in leukaemia. CD1c is a non-polymorphic antigen presenting molecule expressed on the cell surface of leukocytes including malignant B cells. Realizing the cancer immunotherapy potential of this system could provide a step-change in our ability to treat both childhood and adult AML and ALL, particularly for those that relapse. Particularly so as this "DURT" approach reduces Graft vs host disease and it can be applied to the wider population. The authors start by reporting CD1c expression on several human leukaemia and lymphoma subtypes providing good justification for targeting CD1c in these human cancers. The authors then go on to investigate the use of several CD1c restricted TCRs as anti-tumour agents and they show that TCR DN4.99 is highly efficient with regards to being stably expressed on donor T cells but also via its recognition and mediated killing of leukemic cell lines including primary leukemic blasts. DN4.99 transduced T cells are specific as they do not kill normal healthy CD1c expressing monocytes, DCs and B cells which should limit off-target killing in vivo. The authors finish off with elegant pre-clinical mouse in vivo studies that further supports employing this novel approach in cancer immunotherapy. The study is well conducted, with all the required controls in place and the data is convincing. The authors provide the platform for future use if this novel approach in cancer therapy and it is worthy of publication.

I have a few comments:

- Why did the authors choose DN4.99 TCR? It seems to me that the data shown in figure 2 suggests that PZP8A6 is just as good. Also, a better explanation for figure 2g is warranted.
- In figure 5d, I was surprised to see that B cells were not killed by DN4.99 TCR T cells even in the presence of mLPA? Do the authors have an explanation for that
- In the data presented in figure 7, the control was a Mtb specific TCR whereas in figure 6 the authors used untransduced T cells as control. Why did the authors use DL15TCR here as a control and what was the rationale in using these different controls in the in vivo studies?
- In the in vivo studies, you inject mLPA loaded DC into the mice to boost DN4.99 TCR T cells in vivo. As the tumour cells "MOLM-13" expressed CD1c, why did they not sustain the expansion of the T cells? Furthermore, although mice do not express CD1c in the discussion it was suggested that mLPA could potentially be used in patients to sustain the response. Is this feasible as it may lead to off-target killing? Could the authors expand on this?

Minor:

- On line 169, CD4+ one should be CD4+ cells

I have no further comments.

Reviewer #2 (Remarks to the Author):

In this report, M. Consonni et al. describe development of a new form of immunotherapy for CD1c-positive hematologic malignancies. The study builds logically on prior work from this group of authors who discovered that CD1c-positive leukemia cells but not normal leukocytes express high levels of methyl-lysophosphatidic acid (mLPA) and that CD1c-restricted T cells can specifically and selectively kill leukemia cells. As the next step towards clinical application of this discovery, authors validated CD1c expression in large datasets from patients with AML, B-ALL, and DLBCL and performed a systematic evaluation of CD1c/mLPA TCRs. They selected one TCR (DN4.99), which when transduced in T cells enables potent killing of CD1c-positive leukemia lines and primary blasts from patients, but not CD1c-positive normal leukocytes. Authors then demonstrated in three in vivo models that adoptive transfer of DN4.99 TCR-T cells mediate antitumor activity against established leukemia/lymphoma xenografts and that the therapeutic potency can be enhanced by repeat TCR-T cell infusion or by a boost vaccination with mLPA-loaded moDC.

The significance of this work is very high. Clinical applications of conventional HLA-restricted TCRs for cancer immunotherapy are limited due to HLA polymorphism and patient specificity of tumor antigens. CD1c is a non-polymorphic HLA class I-like molecule and therefore, CD1c-specific T cells reactive to a shared CD1c-specific tumor antigen can be universally applied for treating CD1c-

positive leukemia/lymphoma without a risk of GvHD. Moreover, authors demonstrated specificity of DN4.99 to the CD1c/mLPA complex and not to CD1c itself that enables selective targeting of CD1c-positive leukemia/lymphoma cells, sparing normal leukocytes, the only CD1c-expressing tissue in humans. The experiments are exceptionally well designed and executed. The results inform further development of the "off-the-shelf" immunotherapy with adoptively transferred CD1c-restricted TCR-T cells in patients with CD1c-positive hematologic malignancies. Authors also articulated current limitations of their TCR-T therapy in the Discussion section, providing insights for future development.

Reviewer #3 (Remarks to the Author):

In the present study Consonni et al examined the therapeutic potential of primary human T cells engineered to express an mLPA-specific TCR. The choice to use a mLPA/CD1c restricted TCR is of peculiar interest due to the monomorphism of CD1c and the high enrichment of mLPA in tumor cells.

However, in a previous article (Lepore M et al J Exp Med 2014) the same team has already published a set of experiments characterizing mLPA/CD1c-specific T cells clones and their antitumor activity both in vitro and in vivo which limits the novelty of this study.

This paper reinforces the feasibility of targeting this mLPA/CD1c complex but the novelty is limited.

1- How the choice of the data set presented in Fig 1 or Supp Fig 1a was done?

2- Line 103 "One data set (GSE18497)" Is it the only one with data at diagnosis and at relapse or the only one to show comparable levels of CD1 gene expression?

3- Line 109 A A is missing for B-ALL (and Line 112). CD1c seems to be also expressed in T-ALL in Fig1a why did you exclude T-ALL from the potential indications?

4- Why did the authors use CD1c transduced cell lines in vitro and not AML or B-ALL cell lines?

5- Line 136 Authors suggest that CDR3 are important for mLPA recognition due to the absence of P8E3 TCR-JK activation contrary to DN4.99 TCR-JK. However, the P8E3 T cell clone with the same CDR3 is able to recognize mLPA.

6- Line 140 What did the authors mean by stronger reactivity? Affinity or avidity. This could explain the strongest endogenous TCR repression Fig 2a.

7- Line 145 DN4.99 TCR activation induces a strong long lasting (more than 3weeks) but transient down regulation of endogenous TCR. The conclusion that it will minimize the risk of alloreactivity is overstated.

8- Line 148 The relevance of the use of CD62L in the absence of CCR7 to characterize Tscm is low.

9- Line 149 Could you specify the IL-7 and IL-15 doses in IU/mL in the Methods section?

10- Line 161 Why did the authors use only two untransduced cell lines? It should be noted that one of them is a T-ALL (see comment 3).

11- Line 182 Was the experiment performed with autologous or allogenic DN4.99TCR-T cells?

12- Line 185 At the end of the experiment living blasts were still detected despite controlling the E:T ratio limiting the therapeutic potential of this strategy.

13- Could the authors use PDX in place of CD1c transduced cell lines to improve the relevance of these in vivo experiments?

14- Line 227 Statistics are not shown in Supp Fig 2g.

15- Line 240 It seems to me that the two T cell transfers were also performed in fig 7b.

This is an interesting study which may have translational potential. However, several points require further clarification before conclusions can be made.

Point-by-point response to the reviewers' comments

Reviewer #1 (Remarks to the Author):

In the study by Consonni and colleagues, the CD1c mediated anti-tumour immune response is investigated for its therapeutic potential in leukaemia. CD1c is a non-polymorphic antigen presenting molecule expressed on the cell surface of leukocytes including malignant B cells. Realizing the cancer immunotherapy potential of this system could provide a step-change in our ability to treat both childhood and adult AML and ALL, particularly for those that relapse. Particularly so as this “DURT” approach reduces Graft vs host disease and it can be applied to the wider population. The authors start by reporting CD1c expression on several human leukaemia and lymphoma subtypes providing good justification for targeting CD1c in these human cancers. The authors then go on to investigate the use of several CD1c restricted TCRs as anti-tumour agents and they show that TCR DN4.99 is highly efficient with regards to being stably expressed on donor T cells but also via its recognition and mediated killing of leukemic cell lines including primary leukemic blasts. DN4.99 transduced T cells are specific as they do not kill normal healthy CD1c expressing monocytes, DCs and B cells which should limit off-target killing *in vivo*. The authors finish off with elegant pre-clinical mouse *in vivo* studies that further supports employing this novel approach in cancer immunotherapy. The study is well conducted, with all the required controls in place and the data is convincing. The authors provide the platform for future use if this novel approach in cancer therapy and it is worthy of publication.

Reply. We thank the reviewer for remarking that our study is well conducted, convincing and novel, worthy of publication.

1. Why did the authors choose DN4.99 TCR? It seems to me that the data shown in figure 2 suggests that PZP8A6 is just as good. Also, a better explanation for figure 2g is warranted.

Reply. We agree with the reviewer that also the PZP8A6 TCR is interesting. For the current study, we chose DN4.99 TCR as the lead TCR, instead of PZP8A6, because it recognized the cognate mLPA-CD1c complex with a lower EC_{50} (239 pg/ml vs 339 pg/ml), implying a higher functional avidity, and it induced the strongest (Figure 3b) and most sustained (Supplementary Figure 2a) endogenous TCR downregulation in the transduced T cells. These functional characteristics of the DN4.99 TCR provided relevant advantages in terms of expression on T cells and efficacy of target recognition, as well as reducing xenoreactivity in the *in vivo* experiments, compared to the PZP8A6. Figure 3b shows the downregulation of the endogenous TCRs induced by the transduction of the three indicated chimeric TCRs. Shown in dark are the histograms depicting the surface expression of the

endogenous TCRs in the transduced T cells, detected by using anti-human TCR $\alpha\beta$ -specific mAb. The DN4.99 TCR has the greatest ability to downregulate the surface expression of the endogenous TCRs, compared to the other two TCRs. The downregulation of the endogenous TCRs in human T cells upon transduction with recombinant TCRs containing mouse C α / β regions has already been described in the literature. It is explained by a higher affinity of mouse TCR C α / β regions for the human CD3 ζ , compared to the human C regions (Cohen CJ, Cancer Res, 2006). This results in a sequestration of the available CD3 ζ molecules, which are the rate limiting components for the transport of the whole TCR/CD3 complex to the cell membrane (Weissman AM, EMBO J 1989; D'Oro U, J Immunol, 2002), by the transduced chimeric TCRs leading to a reduced membrane transport of the endogenous ones. The higher the surface expression of the transduced chimeric TCRs, the lower the expression of the endogenous ones; that is why the DN4.99 TCR achieves the greatest down regulation of the endogenous molecules. The RFI of the transduced TCR are: DN4.99 TCR RFI 96, PZP8A6 TCR RFI74 and DN4.2 TCR RFI 57. We have clarified these concepts in the Results section (lines 145-148) as follows: *“This is consistent with published reports showing a higher affinity of mouse TCR C α and C β regions for the human CD3 ζ compared to the human C regions¹², resulting in the sequestration of available CD3 ζ molecules, which are the rate limiting components for the transport of the all TCR/CD3 complex to cell membrane^{16,17} by the transduced chimeric TCRs.”*

(Please note that in the revised manuscript figure 2g is now figure 3b).

2. In figure 5d, I was surprised to see that B cells were not killed by DN4.99 TCR-T cells even in the presence of mLPA? Do the authors have an explanation for that.

Reply. The fact that DN4.99 TCR-T cells spare B cells is a positive result in terms of potential safety of ACT with T cells transduced with this TCR, preventing possible side effects originated by their elimination. However, we do not have a ready explanation for this result. We have already observed (Lepore M, J Exp Med, 2014) that B cells were not killed by the original DN4.99 T cell clone, from which the TCR was derived, although they could stimulate low levels of IFN- γ production by the clone when loaded with exogenous mLPA, implying a B cell ability to present the lipid antigen. By contrast, we never detected IFN- γ production by DN4.99 TCR-T cells upon stimulation by either non-loaded or mLPA-loaded B cells (data not shown). One possibility we can envisage is that the T cell clone be more responsive than the transduced T cells to the low stimulation provided by B cells. (Please note that in the revised manuscript figure 5d is now figure 6d).

3. In the data presented in figure 7, the control was a Mtb specific TCR whereas in figure 6 the authors used untransduced T cells as control. Why did the authors use DL15TCR here as a control and what was the rationale in using these different controls in the *in vivo* studies?

Reply. In the data presented in figure 7 (now figure 8 in the revised manuscript) we sought to improve the information obtained by the ACT experiment adding a more rigorous control compared to the non-transduced T cells, consisting of T cells transduced with a TCR sharing CD1c restriction with the DN4.99 TCR, yet specific for an exogenous bacterial lipid antigen, therefore not self-reactive. Indeed, DL15A31 TCR-transduced T cells proved unable to control leukemia progression, confirming the efficacy of the DN4.99 TCR relying on its mLPA-specificity.

4. In the *in vivo* studies, you inject mLPA loaded DC into the mice to boost DN4.99 TCR T cells *in vivo*. As the tumour cells “MOLM-13” expressed CD1c, why did they not sustain the expansion of the T cells?

Reply. We envisage the possibility that MOLM-13 cells may trigger T cell cytotoxicity, but cannot provide sufficient “costimulatory” signals for T cell expansion, unlike DCs. MOLM-13 is, indeed, the most aggressive of the three acute leukemia models that we have utilized *in vivo*, and its poor T cell-stimulatory ability may contribute to this phenotype.

5. Furthermore, although mice do not express CD1c in the discussion it was suggested that mLPA could potentially be used in patients to sustain the response. Is this feasible as it may lead to off-target killing? Could the authors expand on this?

Reply. We thank the reviewer for raising this important issue, which we overlooked in the discussion. CD1c is only expressed by mature leukocytes with APC functions, namely monocytes, dendritic cells, Langerhans cells and B cells. Hence, potential on-target/off-tumor reaction by mLPA-specific T cells would spare essential parenchymatous organs that are negative for CD1c. However, as remarked by the reviewer, it is possible that DN4.99 TCR-T cells may kill a fraction of the CD1c-expressing APCs, at least the myeloid ones, once they have been restimulated following free systemic mLPA injection. Hence, the best clinical approach to minimize the potential issue of depleting endogenous CD1c-expressing APCs remains to boost the adoptively transferred mLPA-specific T cells with donor-derived DCs loaded with the synthetic lipid antigen, as we have achieved in our study. We have accordingly modified the Discussion (lines 342-347) as follows:

*“This vaccine strategy has already been approved for use in patients⁴², and it is a suitable option to improve our ACT strategy. The presence of endogenous CD1c-expressing APCs in man may also support a cell-free formulation of an mLPA vaccine for its direct delivery *in vivo*. However, this*

approach should be carefully evaluated because it may associate with unwanted potential on-target/off-tumor elimination of the endogenous CD1c⁺ APCs presenting mLPA to engineered T cells.”

6. On line 169, CD4+ one should be CD4+ cells.

Reply. “CD4+ one” was replaced with “CD4+ cells” (On line 174 in the revised manuscript).

Reviewer #2 (Remarks to the Author):

In this report, M. Consonni et al. describe development of a new form of immunotherapy for CD1c-positive hematologic malignancies. The study builds logically on prior work from this group of authors who discovered that CD1c-positive leukemia cells but not normal leukocytes express high levels of methyl-lysophosphatidic acid (mLPA) and that CD1c-restricted T cells can specifically and selectively kill leukemia cells. As the next step towards clinical application of this discovery, authors validated CD1c expression in large datasets from patients with AML, B-ALL, and DLBCL and performed a systematic evaluation of CD1c/mLPA TCRs. They selected one TCR (DN4.99), which when transduced in T cells enables potent killing of CD1c-positive leukemia lines and primary blasts from patients, but not CD1c-positive normal leukocytes. Authors then demonstrated in three in vivo models that adoptive transfer of DN4.99 TCR-T cells mediate antitumor activity against established leukemia/lymphoma xenografts and that the therapeutic potency can be enhanced by repeat TCR-T cell infusion or by a boost vaccination with mLPA-loaded moDC.

Reply. We thank the reviewer for properly resuming our previous work and clearly defining the new achievements of the current study.

The significance of this work is very high. Clinical applications of conventional HLA-restricted TCRs for cancer immunotherapy are limited due to HLA polymorphism and patient specificity of tumor antigens. CD1c is a non-polymorphic HLA class I-like molecule and therefore, CD1c-specific T cells reactive to a shared CD1c-specific tumor antigen can be universally applied for treating CD1c-positive leukemia/lymphoma without a risk of GvHD. Moreover, authors demonstrated specificity of DN4.99 to the CD1c/mLPA complex and not to CD1c itself that enables selective targeting of CD1c-positive leukemia/lymphoma cells, sparing normal leukocytes, the only CD1d-expressing tissue in humans. The experiments are exceptionally well designed and executed. The results inform further development of the “off-the-shelf” immunotherapy with adoptively transferred CD1c-restricted TCR-T cells in patients with CD1c-positive hematologic malignancies. Authors also articulated current

limitations of their TCR-T therapy in the Discussion section, providing insights for future development.

Reply. We thank the reviewer for the strong appreciation to our study and for underscoring the high significance and potential advantages of the “off-the-shelf” immunotherapy we defined.

Reviewer #3 (Remarks to the Author):

In the present study Consonni et al examined the therapeutic potential of primary human T cells engineered to express an mLPA-specific TCR. The choice to use a mLPA/CD1c restricted TCR is of peculiar interest due to the monomorphism of CD1c and the high enrichment of mLPA in tumor cells. However, in a previous article (Lepore M et al J Exp Med 2014) the same team has already published a set of experiments characterizing mLPA/CD1c-specific T cells clones and their antitumor activity both *in vitro* and *in vivo* which limits the novelty of this study. This paper reinforces the feasibility of targeting this mLPA/CD1c complex but the novelty is limited.

Reply. We thank the reviewer for highlighting the peculiar interest of targeting mLPA/CD1c for adoptive cell therapy of leukemia. However, we respectfully disagree with the reviewer’s opinion of the limited novelty of the current study, because our previous article (Lepore M, J Exp Med, 2014) mainly described the leukemia antigen-specificity of a series of CD1c self-reactive T cell clones, by identifying the mLPA tumor-lipid antigen, laying the foundation for the development of donor-unrestricted ACT strategy targeting leukemia lipid antigens. In that study we used CD1c self-reactive T cells clones that would be very difficult to isolate and expand from any donor at therapeutic numbers. Hence, our current study substantially extends those previous findings, by defining a therapeutic strategy enabling the generation of unlimited mLPA-CD1c-specific T cells from any donor, based on the rigorous selection of a lead TCR from an extended candidate panel, the optimization of its expression and function in T cells, and the demonstration of its anti-leukemia efficacy *in vitro* and *in vivo*. All these activities were not present in the 2014 study, precluding the definition of the overall therapeutic strategy, and collectively represent substantial advancement from the previous knowledge, as also underscored by the two previous reviewers.

1. How the choice of the data set presented in Fig 1 or Supp Fig 1a was done?

Reply. We thank the reviewer for prompting a clarification of this and the next bioinformatic issues. We performed a survey on public repositories filtering the dataset according to uniformity of platforms, *e.g.* Affymetrix, completeness of available clinical information and with adequate number

of samples for each leukemia type. This clarified information was added to the Results sections (lines 99-101) that now reads:

“To confirm and extend our original findings, we first established the expression levels of all group 1 and 2 CD1 genes in acute leukemia by interrogating 16 publicly-available gene expression datasets from adult and pediatric AML, B-ALL and T cell acute lymphoblastic leukemia (T-ALL) patients (Supplementary Table 1). These datasets were filtered according to uniformity of platforms, e.g. Affymetrix, completeness of available clinical information and with adequate number of samples for each leukemia type.”

2. Line 103 “One data set (GSE18497)” Is it the only one with data at diagnosis and at relapse or the only one to show comparable levels of CD1 gene expression?

Reply. The GSE18497 was the only dataset with matched diagnosis relapse pair samples at our survey. This information was also added to the Results sections (lines 103-104) as follows:

“The GSE18497 dataset, the only one available with matched diagnosis-relapse paired samples, reported comparable levels of CD1 gene expression in B-ALL and T-ALL at diagnosis and at relapse”

3. Line 109 A is missing for B-ALL (and Line 112). CD1c seems to be also expressed in T-ALL in Fig1a why did you exclude T-ALL from the potential indications?

Reply. The typos in line 109 was corrected; the typos in line 112 (114 in the revised version) now reads as: *in acute leukemias and DLBCL.*

The reviewer is right and we have included T-ALL into the list of potential indications for ACT with the DN4.99 TCR, shown in the Results (line 109) and Discussion (line 289) as follows:

Results: *“Thus, CD1c-reactive T cells have the potential to target AML, B-ALL, T-ALL, and DLBC lymphoma expressing CD1c.”*

Discussion: *“In addition, any CD1c-expressing hematological malignancies, such as T-ALL and DLBCL, may also become possible therapeutic targets.”*

4. Why did the authors use CD1c transduced cell lines *in vitro* and not AML or B-ALL cell lines?

Reply. We have indeed utilized three types of malignant targets for the T cell recognition experiments *in vitro* (Figure 4 and 5 in the revised manuscript): 1. Two established naturally CD1c-expressing acute leukemia cell lines (B-ALL CCRF-SB; T-ALL MOLT4), which were available to us; 2. Four established acute leukemia cell lines that were engineered to express CD1c upon gene transduction (AML: THP-1, MOLM-13, K562; B-ALL NAML6); 3. Primary circulating CD1c-expressing AML and B-ALL blasts obtained from the blood of N= 9 leukemia patients at diagnosis (B-ALL 31; AML-

11, AML-28, AML-32, AML-34, AML-42, AML-44, AML-45, and AML-48). This has allowed us to sample a very broad range of target cell types, supporting the recognition ability and specificity of DN4.99 TCR-T cells.

5. Line 136 Authors suggest that CDR3 are important for mLPA recognition due to the absence of P8E3 TCR-JK activation contrary to DN4.99 TCR-JK. However, the P8E3 T cell clone with the same CDR3 is able to recognize mLPA.

Reply. We thank the reviewer for pointing this out, and here we clarify this assumption. The P8E3 T cell clone was originally selected for being CD1c self-reactive, but it was never described in any our previous publication. We did that for this study. For the reviewer’s perusal, we report below one of the several (and reproducible) mLPA-recognition experiments done with the two T cell clones. The P8E3 clone recognizes THP1-CD1c cells loaded with mLPA, although at substantial lower level (as shown by the IFN- γ production) than the DN4.99 clone. Because TCR-transduced Jurkat cells are inherently less responsive to cognate antigen-stimulation than the original T cell clones, from which the TCR has been derived, it is not surprising that P8E3 TCR-JK cells cannot respond to the cognate stimulation, unlike the DN4.99 TCR-JK ones. Together, given that the DN4.99 and P8E3 TCRs share the same V α and V β regions, but different CDR3s, these series of results strongly suggest a role for CDR3 regions in determining the recognition of mLPA.

Recognition of THP-1-CD1c cells by P8E3 and DN4.99 T cell clones. T cells were cultured at 2.5:1 E:T ratio with THP-1-CD1c cells loaded with 1.5 ng/ml of mLPA +/- blocking anti-CD1c mAb. IFN- γ secretion into supernatants was measured by ELISA after 48h of co-culture.

6. Line 140 What did the authors mean by stronger reactivity? Affinity or avidity. This could explain the strongest endogenous TCR repression Fig 2a.

Reply. For stronger reactivity, we mean that the DN4.99 TCR has a higher “functional avidity” compared to the other TCRs, supported by its higher response to lower antigen concentration (EC₅₀). We refer to the established “functional avidity” concept as the measure describing how well a T cell responds *in vitro* to a given concentration of a ligand. T cells with high functional avidity respond to low antigen doses, while T cells with lower functional avidity require higher amounts of antigen to

achieve a similar level of response (reviewed in Viganò S et al, Clin Dev Immunol 2012; Campillo-Davo D et al, Cancers, 2020). As suggested by the reviewer, it is possible that this may also contribute to the “strongest endogenous TCR repression” by the DN4.99 TCR, compared to the other ones. We have rephrased the sentence in the Results line 140 by replacing “reactivity” with “functional avidity”:

“...the DN4.99 TCR exhibited a lower EC_{50} (half maximal effective concentration) than the other two TCRs, and the highest CD69 expression (Fig. 2e), suggesting stronger **functional avidity**^{14,15} upon antigen engagement.”

7. Line 145 DN4.99 TCR activation induces a strong long lasting (more than 3weeks) but transient down regulation of endogenous TCR. The conclusion that it will minimize the risk of alloreactivity is overstated.

Reply. We thank the reviewer for this remark. We have tempered the text (Lines 148-152) as follows:

“The strong downregulation of the endogenous TCRs by the DN4.99 TCR is a characteristic that might improve the safety of adoptive cell therapy (ACT) in allogeneic HSCT, by reducing the expression of potentially alloreactive TCRs implicated in pathogenesis of GvHD, although additional strategies to prevent GvHD by TCR-T cells are warranted for clinical application.”

Furthermore, in the original Discussion we already addressed at length this specific safety issue (Line 308-314): “Nonetheless, additional strategies to prevent GvHD by the TCR-transduced T cells could be envisaged, such as knocking-down endogenous TCR expression using genome-editing approaches³⁴, or transferring the anti-tumor TCR into the unconventional iNKT, MRI- or $\gamma\delta$ -T cell subsets that do not elicit GvHD upon allogeneic transfer because they are restricted to non-polymorphic molecules and express TCRs with strong intra-chain physical constraints, thereby minimizing mispairing with endogenous TCR chains^{35,36}.”

8. Line 148 The relevance of the use of CD62L in the absence of CCR7 to characterize Tscm is low

Reply. As suggested by the reviewer, we repeated the staining of T cells expanded *in vitro* adding anti-CCR7 mAb to confirm the characterization of Tscm cells. As shown in the new supplementary figure 2b-c enclosed below, both *in vitro* expanded DN4.99 TCR-T and non-transduced T cells contain CD45RA⁺CD62L⁺CD95⁺ populations (Tscm cells according to the work published by our coauthor Chiara Bonini Blood 2014; Sci Transl Med 2015, Nature Med 2017) that also express CCR7, at various extent in CD4⁺ and CD8⁺ cell populations, thus confirming the presence of *bona fide* Tscm

populations. The new figure has been substituted in the supplementary material (supplementary Fig.2).

Supplementary Fig. 2. [...] (b-c) Distribution of T cell subpopulations (CCR7⁺CD95⁺ stem cell memory [T_{SCM}], central memory [T_{CM}], effector memory [T_{EM}] and terminal effector [T_{TE}]) within DN4.99 TCR-T cells (b) and non-transduced T cells (c) at day 17 from T cells restimulation with the Dynabeads® human T-Activator CD3/CD28 (Thermofisher) at a 3:1 bead:cell ratio. The flow cytometry analysis was performed labeling the cells with anti-mouse (m)TCRβ-APC/Cy7, CD8-APC (HIT8a), CD62L PE/Cy7 (DREG-56), CD45RA FITC (HI100); CCR7-PE (G043H7), CD95 PE (DX2; all from BioLegend) and CD4-V500 (RPTA-4, BD) mAbs.

9. Line 149 Could you specify the IL-7 and IL-15 doses in IU/mL in the Methods section?

Reply. IL-7 and IL-15 were added to cultures at approximately 22×10^2 IU/ml each. This has been now indicated in Methods (line 440).

10. Line 161 Why did the authors use only two untransduced cell lines? It should be noted that one of them in a T-ALL (see comment 3).

Reply. As already pointed out for question #4, we used *in vitro* a broad range of established acute leukemia cell lines available in the lab, as well as primary circulating leukemia blasts from patients' blood. Only two of these available cell lines turned out to naturally express CD1c, as we assessed by flow cytometry screening, and it was not clear from either the literature, public database or the available cell repositories whether it existed other acute leukemia cell lines that naturally expressed CD1c on the cell membrane. For this reason, we have generated 4 more targets by transducing the CD1c cDNA into the negative cell lines THP-1, MOLM-13, K562 and NAML-6. Of note, three of the CD1c-transduced leukemia cell lines (THP-1, NAML-6, MOLM-13) are also well-established models of *in vivo* human leukemia xenografts in NSG mice, thus providing validated targets for the

ACT experiments. As already replied for question #3, we have included T-ALL in the list of potential targets for our ACT strategy.

11. Line 182 Was the experiment performed with autologous or allogenic DN4.99TCR-T cells?

Reply. The experiment was performed with allogeneic DN4.99 TCR-T cells from healthy donors. The specificity of the DN4.99 TCR recognition is confirmed by its blocking by the addition of anti-CD1c mAb, while the downregulation of the endogenous allogeneic TCRs prevents the potential allorecognition of target cells. We have specified the allogeneic origin of T cells in the Figure 5 legend of the revised manuscript:

“Allogenic CD8⁺/CD4⁺ DN4.99 TCR-T cells were from 3 independent healthy donors.”

12. Line 185 At the end of the experiment living blasts were still detected despite controlling the E:T ratio limiting the therapeutic potential of this strategy.

Reply. The *in vitro* DN4.99 TCR T:primary blast co-cultures were performed for 48h (in the text it was wrongly indicated 72h) to visualize the specificity of killing and, at the same time, to avoid the spontaneous leukemia cell death. During this time, however, T cells may not completely eliminate the primary blasts. A 72 h T:blasts coculture time resulted in the spontaneous leukemia cell death that precluded the interpretation of the killing results.

13. Could the authors use PDX in place of CD1c transduced cell lines to improve the relevance of these *in vivo* experiments?

Reply. Unfortunately, after some attempts, we do not yet have CD1c⁺ leukemia blasts that engraft in NSG mice. However, we are keeping trying to establish usable CD1c⁺ PDX for future experiments. Nevertheless, the three acute leukemia cell lines that we have utilized for the ACT experiments *in vivo* are of broad use in the literature for TCR- and CAR-T cell studies. Hence, we feel confident to have exploited recognized models of human acute leukemia xenografts.

14. Line 227 Statistics are not shown in Supp Fig 2g.

Reply. Statistics was added as suggested.

15. Line 240 It seems to me that the two T cell transfers were also performed in fig 7b.

Reply. The reviewer is right, we apologize for the confusion. The text was modified as follow (line 245):

“Here, we saw that leukemia progression was significantly delayed in mice that received DN4.99 TCR-T cells, compared to the two control groups (Fig. 8b). A second transfer of DN4.99 TCR-T cells, performed 7 days after the first one, further significantly delayed leukemia progression (Fig. 8b), as also shown by reduced frequency of circulating blasts at day +17 from tumor injection (Fig. 8c).”

(Please note that in the revised manuscript figure 7 is now figure 8).

This is an interesting study which may have translational potential. However, several points require further clarification before conclusions can be made.

Reply. We thank the reviewer for finally considering our study interesting and underscoring its translational potential. We hope to have clarified all the points s/he raised and, accordingly, improved our study in the revised version.

REVIEWERS' COMMENTS

Reviewer #1 (Remarks to the Author):

Consonni and colleagues have addressed the comments raised by this reviewer which would hopefully have improved the manuscript.

However, I have one minor comment on the modified text

Minor comment:

Line 148 : transport of the all TCR/CD3 complex to cell membrane

Should be

Line 148: transport of all TCR/CD3 complexes to the cell membrane

I have no further comments.

Reviewer #2 (Remarks to the Author):

Outstanding work!

Reviewer #3 (Remarks to the Author):

The authors have satisfactorily answered the questions raised in the review.

Reviewer #1 (Remarks to the Author):

Consonni and colleagues have addressed the comments raised by this reviewer which would hopefully have improved the manuscript.

However, I have one minor comment on the modified text

We are delighted that the reviewer finds our revised manuscript improved.

Minor comment:

Line 148 : transport of the all TCR/CD3 complex to cell membrane

Should be

Line 148: transport of all TCR/CD3 complexes to the cell membrane

The line has been corrected

Reviewer #2 (Remarks to the Author):

Outstanding work!

We thank the reviewer for appreciating our revised manuscript.

Reviewer #3 (Remarks to the Author):

The authors have satisfactorily answered the questions raised in the review.

We are pleased that the reviewer finds our revision satisfactory.